# QUALITY MEASURES FOR DYNAMIC GRAPH GENERATIVE MODELS

**Ryien Hosseini**[1]**, Filippo Simini**[2]**, Venkatram Vishwanath**[2]**, Rebecca Willett**[1,3,4]**, Henry Hoffmann**[1]

[1]Department of Computer Science, University of Chicago
[2]Leadership Computing Facility, Argonne National Laboratory
[3]Department of Statistics, University of Chicago
[4] NSF-Simons National Institute for Theory and Mathematics in Biology
{ryien,willett,hankhoffmann}@uchicago.edu, {fsimini,venkat}@anl.gov

## ABSTRACT

Deep generative models have recently achieved significant success in modeling graph data, including dynamic graphs, where topology and features evolve over time. However, unlike in vision and natural language domains, evaluating generative models for dynamic graphs is challenging due to the difficulty of visualizing their output, making quantitative metrics essential. In this work, we develop a new quality metric for evaluating generative models of dynamic graphs. Current metrics for dynamic graphs typically involve discretizing the continuous-evolution of graphs into static snapshots and then applying conventional graph similarity measures. This approach has several limitations: (a) it models temporally related events as i.i.d. samples, failing to capture the non-uniform evolution of dynamic graphs; (b) it lacks a unified measure that is sensitive to both features and topology; (c) it fails to provide a scalar metric, requiring multiple metrics without clear superiority; and (d) it requires explicitly instantiating each static snapshot, leading to impractical runtime demands that hinder evaluation at scale. We propose a novel metric based on the *Johnson-Lindenstrauss* lemma, applying random projections directly to dynamic graph data. This results in an expressive, scalar, and application-agnostic measure of dynamic graph similarity that overcomes the limitations of traditional methods. We also provide a comprehensive empirical evaluation of metrics for continuous-time dynamic graphs, demonstrating the effectiveness of our approach compared to existing methods. Our implementation is available at https://github.com/ryienh/jl-metric.

## 1 INTRODUCTION

Recent research in the generative graph domain has increasingly focused on *dynamic* graphs, or graphs whose topologies and features change over time. Dynamic graph generation has applications in diverse areas, including social network analysis (Patel et al., 2018; Aldhaheri & Lee, 2017), biology (Mathur & Chakka, 2020; Choudhury & Chowdhury, 2018), and financial fraud detection (Rajput & Singh, 2022; Wang et al., 2021). Unlike image or natural language data, graph data can be challenging to visualize, making subjective evaluation of generative samples difficult. Consequently, quality metrics for assessing these generative models are vitally important.

Current metrics for dynamic graph generative models (DGGMs) generally rely on computing (static) graph statistics, such as node degree, number of connected components, or edge entropy, and aggregating them in a way that facilitates scalar comparison between synthetically generated and ground truth graphs. These metrics can be can be categorized into *discrete* and *continuous*. Discrete metrics treat a dynamic graph as a series of static graph snapshots, first calculating static graph statistics for each snapshot and then aggregating these values using a summary statistic, such as mean, or, more rigorously, a distance metric for high-dimensional distributions such as Maximum Mean Discrepancy (MMD) (Gretton et al., 2006). Such a method treats each snapshot of the evolving graph as independently and identically distributed (i.i.d.) and thus fails to capture the temporal dependencies often present in real world graphs (Sizemore & Bassett, 2018). Despite this limitation, we find that

discrete metrics are commonly used in the DGGM literature (See Section 2.2). Continuous graph metrics attempt to remedy this issue by directly incorporating temporal dependencies into the evaluation process, such as by measuring the rate of change of graph attributes over time (Sizemore & Bassett, 2018; Nicosia et al., 2013). However, both discrete and continuous approaches suffer from limitations: First, these metrics focus exclusively on the *topological* properties of graphs, neglecting the equally important evolution of node and edge *features*. While standard statistical methods such as Jensen-Shannon Divergence (Fuglede & Topsoe, 2004) can measure the fidelity of generated features to the ground truth, capturing dependencies—both between individual features and between feature evolution and topology—remains non-trivial, yet essential because these dependencies are precisely what make dynamic graphs interesting and important. Second, these topological metrics depend on a variety of graph statistics, making the evaluation of DGGMs challenging: The ranking of generative models can vary depending on the specific statistic used, and there is no clear, application-domain-agnostic method for determining which metric should take precedence. Finally, since most of these metrics require explicit construction of graph snapshots at each interaction time, their memory and runtime demands can quickly become impractical at scale.

To remedy similar issues in other domains, untrained neural networks have recently been used in image generative model evaluation (Xu et al., 2018; Naeem et al., 2020) and static graph information extraction (Kipf & Welling, 2016b; Morris et al., 2019). Building on these advancements, Thompson et al., 2022 introduces the use of such a *random network* as a viable approach for generative model evaluation in the static graph domain. They demonstrate that a random graph neural network can be used as a feature extractor to embed synthetic and ground truth graphs. These embeddings are then compared using standard distance metrics such as Fréchet Distance (Heusel et al., 2017b) or MMD in order to produce a scalar metric. Though this work stops short of theorizing *why* such a method may be reasonable, they show empirically that these *neural network-based metrics* effectively capture both graph topology and features, yielding a unified scalar score. However, because this approach is designed for static graphs, it does not account for the temporal evolution crucial to dynamic graphs.

Inspired by this work in the image and static graph domain, we develop a similar approach for continuous time dynamic graphs (CTDGs). Specifically, we first argue that the success of random networks as feature extractors may be due to the famed Johnson-Lindenstrauss (JL) lemma (Johnson & Lindenstrauss, 1984), which proves that data transformation via random linear maps approximately preserves similarity with high probability. Leveraging this insight, we introduce a novel DGGM metric by combining random feature extraction via the JL lemma with standard vector similarity measures. Another major contribution of this work is the first comprehensive empirical evaluation of DGGM metrics. Through extensive experiments, we assess the metrics' ability to meet key properties established as essential for generative metrics (Thompson et al., 2022; Xu et al., 2018): *fidelity*, *diversity*, *sample efficiency*, and *computational efficiency*. Finally, we show that our proposed metric addresses limitations of existing methods, providing a more robust solution for evaluating DGGMs.

## 2 BACKGROUND AND RELATED WORK

The evaluation of generative models in terms of their alignment to the training distribution can be classified into two categories: likelihood-based methods (Heusel et al., 2017b) and sample-based methods (Theis et al., 2015). Likelihood-based methods are often intractable (Theis et al., 2015), including for autoregressive graph generative models (Chen et al., 2021; Thompson et al., 2022). Therefore, we follow recent analogous work in the static graph domain by Thompson et al., 2022 and focus on sample-based methods. Specifically, sample-based metrics estimate a distance $\rho$ between real and synthetic distributions $P_r$ and $P_g$ using empirical samples $\mathbb{S}_r = \{\mathbf{x}_1^r, ..., \mathbf{x}_m^r\} \sim P_r$ and $\mathbb{S}_g = \{\mathbf{x}_1^g, ..., \mathbf{x}_n^g\} \sim P_g$ [1]. Thus, we have the distance estimator $\hat{\rho}(\mathbb{S}_g, \mathbb{S}_r) \approx \rho(P_g, P_r)$. Here, *function descriptor* $\mathbf{x}_i$ is a feature vector that characterizes a single sample. Next, we turn to graph representation and demonstrate how such sample-based measures can be applied to dynamic graphs.

### 2.1 CONTINUOUS TIME DYNAMIC GRAPHS

Early work in dynamic graph generation focused on learning distributions over *discrete-time dynamic graphs* (DTDGs). A DTDG consists of a sequence static graphs which are equally spaced

---

[1]For ease of comparison, we adopt notation consistent with Thompson et al. (2022) wherever possible.

in time. While DTDGs are useful in applications where data is captured at regular time intervals (Kazemi, 2022), *continuous-time dynamic graphs* (CTDGs) generalize DTDGs, offering greater flexibility and efficiency. The literature has thus progressively shifted towards CTDG generation due to their ability to capture temporal dependencies more accurately and to handle data with irregular time intervals. This work concentrates on metrics for evaluating CTDGs. A CTDG is represented as a sequence of timestamped events[2], where each event is a change in graph topology (e.g., node or edge creation or deletion) or change in features. Formally, given initial graph state $\mathcal{G}^0$, a set of nodes $\mathcal{V} = \{1, ..., z\}$, and sequence length $k$, our CTDG sequence is:

$$\mathcal{G} = \{c(t_1), c(t_2), ..., c(t_k)\}, \text{where } c(t_i) = (src, dst, t_i, \mathbf{e}_{src,dst}(t_i)) \tag{1}$$

Each event $c \in \mathcal{G}$ parameterized by timestamp $t_i$ is characterized by source and destination nodes of the event $src, dst \in \mathcal{V}$ and $\mathbf{e}_{src,dst}(t_i)$, a directed edge between $src$ and $dst$ at time $t_i$ and that is represented by its feature vector. Thus, unlike DTDGs, an evolving graph with $k$ events can be compactly represented with sequence length $\Theta(|k|)$. An *induced static graph*, or static snapshot, $\mathcal{G}^\tau$ at $t = \tau$ can be explicitly computed by sequentially updating $\mathcal{G}^0$ with events up to time $\tau$. Thus, given temporal resolution $\phi$ we can discretize $\mathcal{G}^\tau$ as $\mathcal{G}^{\text{discrete}} = \{\mathcal{G}^{(0)}, \mathcal{G}^{(\phi)}, \mathcal{G}^{(2\phi)}, \dots, \mathcal{G}^{(\lfloor \tau/\phi \rfloor/\phi)}\}$.

Most generative models for CTDGs, including the aforementioned baselines, are trained on a single ground-truth graph consisting of many events. Implicitly, these methods assume that the covariance between events decays rapidly as the time interval between them increases and their joint probability distribution does not change when all the events are shifted in time, akin to the properties of a wide-sense stationary process. This assumption allows models to treat different temporal segments of the CTDG as approximately independent, enabling pattern extraction and generalization from a single graph. For consistency with existing literature, our empirical evaluation focuses on a model's ability to measure similarity between a generated and ground-truth CTDG in this *single graph, many interactions* setting. However, our methodology is sufficiently general to extend to settings involving multiple CTDGs. In such cases, we assume the different CTDGs are i.i.d., allowing for straightforward aggregation of the scalar metric across graphs using simple distance measures like MMD. This flexibility ensures that our metric can also be used to compare sets of dynamic graphs.

When using sample-based metrics, the goal is to create a function descriptor $\mathbf{x}_i$ for each *event* or *snapshot* in the CTDG, forming an empirical sample $\mathbb{S} = \{\mathbf{x}_1, ..., \mathbf{x}_k\}$. Individual events $c(t) \in \mathcal{G}$ or induced static graphs $G^{t/\phi} \in \mathcal{G}^{\text{discrete}}$ are analogous to graphs in the static graph domain. Unlike static graphs, where each graph (and thus each $\mathbf{x}_i$) is assumed i.i.d., our function descriptor $\mathbf{x}_i$ or distance estimator $\hat{\rho}$ must account for temporal dependencies between events or snapshots.

## 2.2 CURRENT METRICS FOR DGMMS

We now overview common choices for function descriptor $\mathbf{x}_i$ and distance estimator $\hat{\rho}$. To do so, we summarize metrics used for four popular CTDG generative models: TagGen (Zhou et al., 2020), TIGGER (Gupta et al., 2022), Dymond (Zeno et al., 2021), and TG-GAN (Zhang et al., 2021).

The most common choice of function descriptor $\mathbf{x}_i$ is based on static graph topology. Specifically, the ground truth graph $\mathcal{G}_r$ and the synthetic graph $\mathcal{G}_s$ are discretized into a set of induced static graphs, $\mathcal{G}_r^{\text{discrete}}$ and $\mathcal{G}_s^{\text{discrete}}$, using the method described in Section 2.1. For each graph snapshot $G_i \in \mathcal{G}^{\text{discrete}}$, scalar metrics such as average node degree, number of connected components, and edge entropy are calculated. Thus, $\mathbb{S}_r$ and $\mathbb{S}_g$ are sets of scalar values representing a specific static graph statistic. All aforementioned baselines evaluate using such static approaches. Such function descriptor choices may lack the expressiveness needed to capture continuous dynamic behavior (Sizemore & Bassett, 2018). To address this, Dymond and TG-GAN incorporate additional metrics that attempt to capture temporal dependencies and finer-grained dynamics in the evolution of graph structures. Specifically, they develop so-called *node behavior metrics* wherein function descriptor $\mathbf{x}_i$ is based on nodes $\mathcal{V} = \{1, ..., z\}$ rather than induced static graphs. These function descriptors are calculated using classical statistical measures such as activity rate and degree distribution. A comprehensive summary and discussion of all graph statistics reviewed is provided in Appendix A.

---

[2]Events are also referred to as *observations* or *contacts* in the literature.

As discussed above, given real and generated function descriptor sequences $\mathbb{S}_r = \{\mathbf{x}_1^r, ..., \mathbf{x}_m^r\} \sim P_r$ and $\mathbb{S}_g = \{\mathbf{x}_1^g, ..., \mathbf{x}_n^g\} \sim P_g$, we require distance estimator $\hat{\rho}(\mathbb{S}_g, \mathbb{S}_r) \approx \rho(P_g, P_r)$. TagGen and TIGGER limit evaluation to graphs with the same number of snapshots and use mean and median absolute error between $\mathbb{S}_r$ and $\mathbb{S}_g$. Dymond and TG-GAN use Kolmogorov-Smirnov (KS-test) (Massey Jr, 1951) and Maximum Mean Discrepancy (MMD) (Gretton et al., 2006), respectively. Critically, these choices of $\hat{\rho}$ assume an i.i.d. relationship between the function descriptors in $\mathbb{S}_r$ and $\mathbb{S}_g$. However, for the function descriptors described, this assumption is unlikely to hold, as temporal dependencies between snapshots are often present, challenging the validity of these estimators.

Additionally, the function descriptors found in the literature are limited to graph topology and fail to incorporate node or edge features. As discussed in Section 1, a wide range of distance measures, such as Jensen-Shannon divergence (Fuglede & Topsoe, 2004), can be used to evaluate the fidelity of generated features. However, these measures are unable to capture dependencies between the topology and the evolution of features. Thus, current metrics are not expressive enough to model temporal dependencies, topological-feature interactions, and the intricate dynamics observed in real-world graphs. Without a method that can capture these complex relationships, existing approaches remain insufficient for fully assessing the sample quality of DGGMs.

## 2.3 UNTRAINED NEURAL NETWORKS AS FEATURE EXTRACTORS

Recent work in the vision and static graph domains has sought to address similar issues in developing metrics for evaluating generative models. Drawing inspiration from the use of pretrained convolutional neural networks (CNNs) to derive function descriptors $\mathbf{x}_i$ for sample-based metrics (e.g., Bińkowski et al. (2018); Heusel et al. (2017a)), recent work (Xu et al., 2018; Naeem et al., 2020) explores applying untrained neural networks, or *random networks*, for the same task. They find that function descriptors derived from forward-propagating data through a random CNN yield metrics of comparable quality to those obtained from a trained CNN across several choices of distance estimator $\hat{\rho}$. Similarly, Thompson et al. (2022) explores the use of random graph neural networks (GNNs) to produce metrics for static graph data, showing that this method provides more expressive and computationally efficient metrics than those based on classical graph descriptors. In this work, we first investigate why these approaches work, hypothesizing a connection to orthogonal random projections, and then propose a new method for evaluating dynamic graphs based on these insights.

## 2.4 SCORING GENERATIVE METRICS

A major contribution of our work is an empirical study of the quality of various sample-based metrics for CTDGs. Here, we enumerate the established attributes of a good quality metric and detail how such attributes are measured for metrics in the vision and static graph domain. In Section 4, we propose an approach for measuring these attributes for CTDGs.

An expressive metric should capture the *fidelity* of generated samples, meaning it can distinguish between empirical samples $\mathbb{S}_r$ and $\mathbb{S}_g$ drawn from different distributions, with the empirical distance measure $\hat{\rho}$ changing monotonically with the dissimilarity between the two data sets. Xu et al. (2018); O'Bray et al. (2021); Thompson et al. (2022) empirically study this property for images and static graphs using sensitivity analysis. In these studies, a *reference* sample $\mathbb{S}_r$ and a *generated* sample $\mathbb{S}_g$ are initially calculated from identical real-world datasets $\mathcal{D}_r$ and $\mathcal{D}_g$. Then, samples in $\mathcal{D}_g$ are gradually replaced with data from a different distribution (e.g., random data or data from a generative model) and $\mathbb{S}_g$ is re-calculated, parameterizing $\mathbb{S}_g$ as $\mathbb{S}_g(p)$ and $\mathcal{D}_g$ as $\mathcal{D}_g(p)$, where $p$ is the probability of real data replacement in $\mathcal{D}_g$. Thompson et al. (2022) also explores edge perturbation in static graphs, where edges are rewired with probability $p$. The metric response is then studied. Xu et al. (2018) does so subjectively. O'Bray et al. (2021) quantifies the response by reporting the Pearson correlation coefficient between $p$ and $\hat{\rho}(\mathbb{S}_r, \mathbb{S}_g(p))$. Recognizing that Pearson correlation favors linear relationships, Thompson et al. (2022) instead reports the Spearman rank correlation coefficient. We adopt this approach and report Spearman rank correlation in our analysis.

An expressive metric should also capture the *diversity* of generated samples, ensuring that dataset $\mathcal{D}_g$ contains samples from different regions of the probability mass of reference distribution $P_r$, rather than being confined to a limited subset of it. This is particularly crucial in real-world scenarios where the underlying distribution is multi-modal. In generative models, Xu et al. (2018) identifies two common failure points: (1) *mode dropping*, where some modes of $P_r$ are underrepresented or

ignored by the generative model, and (2) *mode collapse*, where there is insufficient diversity within the modes. Thus, a quality metric should be sensitive to such phenomena. To detect such issues, Xu et al. (2018) and Thompson et al. (2022) again employ sensitivity analysis. Data is first clustered into $k$ modes using *k-means* or *affinity propagation* Frey & Dueck (2007). Starting with identical $\mathcal{D}_r$ and $\mathcal{D}_g$, samples in $\mathcal{D}_g$ are progressively replaced with their cluster centers to measure mode collapse. Thus, we can again calculate $\mathbb{S}_r$ and $\mathbb{S}_g(p)$ from $\mathcal{D}_r$ and $\mathcal{D}_g(p)$, where $p$ indicates the probability of replaced data in $\mathcal{D}_g$. To simulate mode dropping, modes are gradually removed from $\mathcal{D}_g(p)$ and data from other modes is duplicated to keep $|\mathcal{D}_g(p)|$ and thus $|\mathbb{S}_g(p)|$, constant. The metric's response is again quantified using the Spearman rank correlation between $p$ and $\hat{\rho}(\mathbb{S}_r, \mathbb{S}_g(p))$.

A quality metric should also be *sample efficient*. That is, it should be able to discriminate between distinct distributions $P_r$ and $P_g$ even when $\lambda = \min(|\mathcal{D}_r|, |\mathcal{D}_g|)$ is small. Thompson et al. (2022) evaluates the sample efficiency of a metric by creating two disjoint subsets $\mathcal{D}'_r$ and $\mathcal{D}''_r$ from real-world data $\mathcal{D}_r$, and one subset $\mathcal{D}'_g$ from random static graph distribution $P_g$, with $|\mathcal{D}'_r| = |\mathcal{D}''_r| = |\mathcal{D}'_g| = \lambda$ and therefore corresponding function descriptors $|\mathbb{S}'_r| = |\mathbb{S}''_r| = |\mathbb{S}'_g|$. The sample efficiency is measured as the smallest $\lambda$ for which the metric $\hat{\rho}(\mathbb{S}'_r, \mathbb{S}''_r) < \hat{\rho}(\mathbb{S}'_r, \mathbb{S}'_g)$, indicating the number of samples needed for the metric to discrimate between $P_r$ and $P_g$.

Finally, a quality metric must be *computationally efficient* to allow for repeated use during tasks like model training and hyperparameter optimization. Computational efficiency is typically assessed by measuring runtime performance on standard computing hardware.

## 3 A JOHNSON-LINDENSTRAUSS APPROACH TO VALIDATING DGGMS

Though the random network-based metrics described in Section 2.3 have found widespread use in the image and static graph domains, these works stop short of explaining *why* such methods may be effective. We argue that their success may be partially attributed to the famed Johnson-Lindenstrauss (JL) lemma (Johnson & Lindenstrauss, 1984; Dasgupta & Gupta, 2003), which posits that random orthogonal linear projections approximately preserve data similarity with high probability. Notably, under mild conditions a randomly initialized fully connected neural network (FCNN) layer with identity activation can be viewed as a direct instance of a JL projection (Nachum et al., 2021). Moreover, recent studies on neural network initialization have shown that FCNNs with ReLU activation and convolutional neural networks (CNNs) modify the JL result in a predictable way, where data similarity is preserved but scaled by a derivable contraction factor (Daniely et al., 2016; Giryes et al., 2016; Nachum et al., 2021). This theoretical foundation may explain the success of random CNN networks as feature extractors in the image domain. While no formal theoretical extension of the JL lemma to the static graph domain has been established, several classes of graph neural networks (GNNs) (e.g., Defferrard et al. (2016); Kipf & Welling (2016a); Wu et al. (2019)) have been rigorously shown to generalize the convolution operation to non-Euclidean structures. Given that the core principle of the JL lemma applies to preserving distances under random projection, we posit that random GNN layers can preserve graph topology and features in a manner analogous to how CNNs preserve data similarity in Euclidean space. This suggests a plausible extension of the lemma's applicability to the graph domain, where the effectiveness of random GNNs as feature extractors may stem from the same underlying principles.

Notably, the distance preservation property, and therefore embedding quality, of the Johnson-Lindenstrauss lemma does not depend on the dimensionality of the original data, leading to widespread use in dimensionality reduction tasks (Bingham & Mannila, 2001; Choromanski et al., 2017), wherein data in $\mathbb{R}^N$ is embedded into a lower-dimensional space $\mathbb{R}^n$, $n < N$. Namely, the lemma states that given distortion factor $0 < \epsilon < 1$, dataset $X = \{a_1, a_2, \ldots, a_q\}, a_i \in \mathbb{R}^N$, and embedding dimensionality $n > 8(\ln q)/\epsilon^2$, there is a linear map $f : \mathbb{R}^N \to \mathbb{R}^n$ such that:

$$(1 - \epsilon)\|a - b\|^2 \leq \|f(a) - f(b)\|^2 \leq (1 + \epsilon)\|a - b\|^2 \tag{2}$$

for all $a, b \in X$, where early proofs of the lemma show that linear map $f : \mathbb{R}^N \to \mathbb{R}^n$ can be a random orthogonal projection from $\mathbb{R}^N$ to subspace $\mathbb{R}^n$.

We argue that this lack of dependence of embedding quality on $N$ is a crucial characteristic that makes Johnson-Lindenstrauss embeddings an effective method for transforming the variable-length node interactions in a CTDG (Equation 1) into a fixed-dimensional vector space. Unlike traditional applications of the JL lemma, where the goal is to reduce dimensionality, a key insight in our work

is to use it to transform data of *varying dimensionality* into a consistent dimensional representation. We now turn our attention to creating a Johnson-Linedenstrauss based metric for CTDGs.

Starting from our CTDG representation in Equation 1, we construct an alternative representation $\tilde{\mathcal{G}}$ at time $\tau$ as a sequence of nodes $\mathcal{V} = \{\mathbf{v}_1, \ldots, \mathbf{v}_z\}$, where each node is characterized by the time-ordered concatenation of events it participated in:

$$\mathbf{v}_j = \{\tilde{c}(t_1)\|\tilde{c}(t_2)\|\ldots\|\tilde{c}(t_{m_j})\},$$

where $m_j$ is the number of events involving node $j$ up to time $\tau$. Since each $\mathbf{v}_j \in \mathcal{V}$ only contains events involving node $j$, the node identifier (either src or dst) is redundant, and we simplify the event representation $c(t_i)$ from Equation 1 to $\tilde{c}(t_i)$, dropping the node identifier and capturing only the timestamp and the associated event features:[3]

$$\tilde{c}(t_i) = (t_i, \mathbf{e}_{\text{src,dst}}(t_i)),$$

As nodes generally participate in different numbers of interactions, the resulting vectors $\mathbf{v}_j$ will have variable lengths, $|\mathbf{v}_j| = m_j \cdot |\tilde{c}(t_i)|$.

However, since the JL embedding quality is agnostic to vector length, we can apply random projections to map each $\mathbf{v}_j$ into a consistent-dimensional space $\mathbb{R}^n$. To achieve this, we instantiate a random projection matrix $W_1^{M \times n}$, where $M = \max_j(|\mathbf{v}_j|)$. We then apply $W_1^{M \times n}$ to each vector $\mathbf{v}_j$, adjusting for variable lengths by ignoring unused rows of the matrix where necessary. This process yields fixed-dimensional function descriptors for each node, $\{\tilde{\mathbf{x}}_1, \ldots, \tilde{\mathbf{x}}_z\}$, where each $\mathbf{x}_j$ characterizes a node $\mathbf{v}_j$ and together they provide a representation for $\mathcal{G}$.

Finally, we note that different CTDGs will generally have different numbers of nodes, resulting in a varying number of function descriptors. To facilitate comparison between CTDGs, we apply an additional transformation that maps the set of node embeddings into a consistent dimensional space. Specifically, for a set of CTDGs $\mathcal{G}_1, \ldots, \mathcal{G}_k$, each with $z_i$ nodes, we instantiate a second random projection matrix $W_2^{Z \times o}$, where $Z = \max(z_1, \ldots, z_k)$ is the maximum number of nodes across the CTDGs, and $o$ is the desired number of function descriptors for each $\mathcal{G}_i$. This transformation yields a consistent representation of each $\mathcal{G}$ as $\tilde{\mathcal{G}} = \{\mathbf{x}_1, \ldots, \mathbf{x}_o\}$, where each $\mathbf{x}_i \in \mathbb{R}^n$. This representation now has consistent dimensionality $\tilde{\mathcal{G}}_i \in \mathbb{R}^{n \times o}$ for all $\mathcal{G}_1, \ldots, \mathcal{G}_k$. Unlike the sequences of function descriptors discussed in Section 2 which each represents individual graph snapshots, $\tilde{\mathcal{G}}_i$ is a matrix representation of the entirety of $\mathcal{G}_i$. Thus, $\mathcal{G}_1, \ldots, \mathcal{G}_k$ can be compared directly using familiar notions of similarity such as cosine distance, i.e. given any pair $\mathcal{G}_r \sim P_r$ and $\mathcal{G}_g \sim P_g$, we have:

$$\rho(P_r, P_g) \approx \hat{\rho}(\tilde{\mathcal{G}}_g, \tilde{\mathcal{G}}_r) = 1 - \frac{\langle \tilde{\mathcal{G}}_g, \tilde{\mathcal{G}}_r \rangle_F}{\|\tilde{\mathcal{G}}_g\|_F \|\tilde{\mathcal{G}}_r\|_F},$$

where $\|\cdot\|_F$ is the Frobenius norm and $\langle \cdot, \cdot \rangle_F$ is the Frobenious inner product. Due to the implicit assumption that the correlation between events decreases rapidly with the time between the events (Section 2.1), this distance estimate between distributions can be accurate even with a single sample from each distribution, as is common in the CTDG literature.

While early works using Johnson-Lindenstrauss transforms employed random orthogonal projection matrices, our approach may use a *structured random matrix* (SRM) discussed by Choromanski et al. (2017), which combines a normalized Hadamard matrix with a random Rademacher diagonal matrix (Appendix B for details). This method has been shown to outperform standard orthogonal projection matrices in dimensionality reduction tasks. By using such SRMs for $W_1^{M \times n}$ and $W_2^{Z \times o}$, we also achieve significant computational and memory benefits: we avoid explicitly instantiating the matrices, reducing storage requirements to $O(M)$ and $O(Z)$ for $W_1$ and $W_2$, respectively. Moreover, matrix-vector multiplication can be performed in $O(M \log M)$ and $O(Z \log Z)$ time, without additional memory overhead. This improved complexity is particularly advantageous when dealing with data of varying dimensionality, such as cases where $\max(|\mathbf{v}_j|) \gg \text{mean}(|\mathbf{v}_j|)$.

This new *JL-metric* addresses the shortcomings of classical CTDG metrics described in Section 2.2:

**Assumption of i.i.d. relationships:** Many classical metrics assume an i.i.d. relationship between graph snapshots, which we have demonstrated is not true in practice. In contrast, our method makes

---

[3]In practice, we also normalize timestamps and event features using standard techniques. See Appendix B.

no such assumption. The transformation $W_1^{M \times n}$ linearly combines the events that constitute $\mathbf{v}_i$, effectively capturing dependencies between events and ensuring that interactions within each node are not treated as independent. A similar effect is achieved by the transformation $W_2^{N \times o}$, which combines node-level descriptors to produce the final metric, preserving dependencies across nodes.

**Joint modeling of topology and features:** Current metrics typically fail to jointly model topological and feature properties of the graph. In contrast, the metric produced by our method is sensitive to both topological and feature changes, as is discussed above and demonstrated in our experiments.

**Unified metric:** Classical metrics often sensitive to only specific graph properties, resulting in multiple metrics with no clear hierarchy for choosing the best model when conflicts arise. Conversely, our method provides a single scalar metric, offering a more comprehensive assessment.

**Efficiency in memory and runtime:** Many classical metrics require explicit instantiation of discrete graph snapshots, resulting in impractical memory and runtime demands for large graphs. In contrast, our method requires only $O(M + Z)$ memory and $O(\max(M, Z) \log \max(M, Z))$ runtime.

## 4 EXPERIMENTS

We now turn to an empirical evaluation of CTDG metrics, comparing several existing metrics with our proposed Johnson-Lindenstrauss-based metric (*JL-Metric*). The comparison focuses on common *desiderata* of generative metrics outlined in Section 2.4: fidelity, diversity, sample efficiency, and computational efficiency. We follow the experimental framework of studies for metrics in the image (Xu et al., 2018) and static graph (Thompson et al., 2022) domains, adapting them for CTDGs.

**Metric Baselines:** In addition to the JL-Metric introduced in Section 3, we evaluate the following topological commonly used function descriptors $\mathbf{x}_i$ from Section 2.2: average node degree, largest connected component (LCC), number of components (NC), power law exponent (PLE) (static metrics), and average node activity rate (node behavior metric). These metrics are computed using standard definitions, which are also provided in Appendix A. A more comprehensive review of function descriptors and motivation for selecting these descriptors can also be found in Appendix A. As described in Section 2.2, static metrics are evaluated for a specific temporal resolution. We evaluate each static metric at the Nyquist rate, the minimum temporal resolution required to create snapshots without information loss. In practice, these metrics can be evaluated at a lower resolution, but this often comes at the cost of reduced accuracy and loss of important temporal details.

For choice of distance estimator $\hat{\rho}$, we use Maximum Mean Discrepancy (MMD) (Gretton et al., 2006) and Kolmogorov-Smirnov (KS) (Massey Jr, 1951) for all metric baselines. Cosine distance is used for our JL-Metric, as motivated in Section 3. Lastly, we evaluate feature-based metrics, where features are compared independently of topology, using Kullback-Leibler divergence (Kullback, 1951), Jensen-Shannon divergence (Fuglede & Topsoe, 2004), as well as KS and MMD.

**Datasets:** We evaluate each metric on four real-world datasets and one synthetic dataset. The real-world datasets are adapted from user interactions on online platforms: Reddit, Wikipedia, LastFM, and MOOC. We use a subset of these data (details in Appendix C), which were originally introduced by Jodie (Kumar et al., 2019) and have become standard CTDG benchmarks. We note that the LastFM dataset does not include event features. The synthetic dataset, $\mathcal{G}_{\text{grid}}$, is a grid-like graph where each node is connected to its neighboring nodes at regular time intervals $t = \{t_1, \ldots, t_n\}$. The event feature $\mathbf{e}_{\text{src,dst}}(t_i)$ is heuristically generated as a function of the timestamp and the source and destination node IDs, i.e., $\mathbf{e}_{\text{src,dst}}(t_i) = h(\text{src}, \text{dst}, t_i)$, with $h$ detailed in Appendix C. This design explicitly introduces temporal and topological dependencies to the event features.

**Experimental Setup:** Building on analogous work from the image and static graph domains (see Section 2.4), we evaluate each metric's ability to capture fidelity (Section 4.1) and diversity (Section 4.2) using sensitivity analysis. In each experiment, we compare a real CTDG $\mathcal{G}_r \sim P_r$ with a perturbed CTDG $\mathcal{G}_g \sim P_g$, the latter serving as a proxy for a DGGM-generated graph. Initially, $\mathcal{G}_r = \mathcal{G}_g$, so $P_r = P_g$, meaning two identical copies of the real CTDG are instantiated. The events in $\mathcal{G}_g$ are then subjected to increasing perturbation probability $p \in [0, 1]$, quantifying the dissimilarity between $P_g$ and $P_r$. We compare $\mathcal{G}_r$ and $\mathcal{G}_g(p)$ for several $p$, computing descriptors $\mathbb{S}_r$ and $\mathbb{S}_g(p)$ for each graph and assessing each metric's response by calculating the Spearman rank correlation between metric score $\hat{\rho}(\mathbb{S}_r, \mathbb{S}_g(p))$ and perturbation probability $p$.

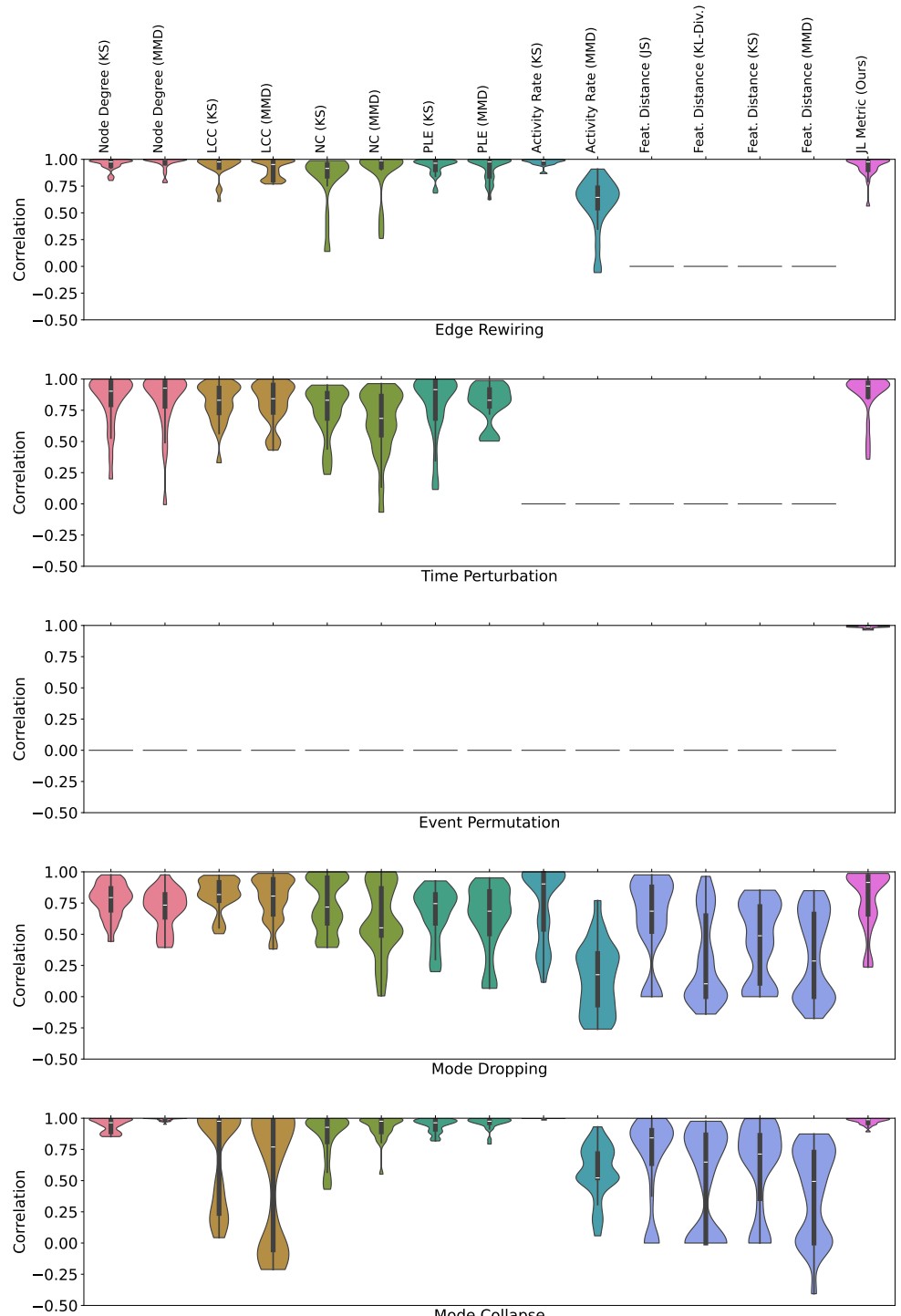

Figure 1: Distributions of Spearman rank correlations across all datasets and random seeds. Correlation is measured between metric response $\hat{\rho}(\mathcal{G}_r, \mathcal{G}_g(p))$ and perturbation probability $p$. Each subplot corresponds to a distinct perturbation scheme, as outlined in Sections 4.1 and 4.2. White lines represent median values, and thick black bars indicate the interquartile range. For classical metrics, colors are mapped based on the function descriptor, independent of the distance estimator.

The experiments for fidelity and diversity follow this common setup, differing only in the types of perturbation applied. Each metric is evaluated across 10 random seeds, which affect both the weights $W_1$ and $W_2$ in the JL-based metrics and the applied perturbations. We report the average rank correlation across all 10 seeds and 5 datasets, with dataset-specific results provided in Appendix E. Additionally, the hyperparameters $n$ and $o$ for the JL-based metric are selected via grid search, as detailed in Appendix D. The experiments on sample and computational efficiency (Section 4.3) are not evaluated via sensitivity analysis; their setup is detailed in the respective section.

## 4.1 MEASURING FIDELITY

The primary goal of a metric is to assess fidelity, or how closely a generated $\mathcal{G}_g$ resembles the ground truth $\mathcal{G}_r$. To evaluate a metric's fidelity, we apply three perturbations: *edge rewiring*, where edges are rewired to a new destination node with probability $p$, altering the topology; *time perturbation*, where, with probability $p$, a timestamp $t_i$ is replaced by a uniformly selected one $t_{\text{rand}} \sim \text{Unif}(t_{i-1}, t_{i+1})$, altering temporal relationships while preserving event order; and *event permutation*, where event feature $\mathbf{e}_{\text{src,dst}}(t_i)$ is replaced by that of another randomly selected event with probability $p$, modifying the feature-topology relationship but preserving the features themselves.

**Results:** Figure 1 (top 3 rows) presents violin plots showing the distributions of correlations across all datasets and random seeds, for all tested metrics. Median values are summarized in Table 1. These results show that commonly used topology-based metrics are comparatively sensitive to edge rewiring, which only alters graph topology. However, they are less sensitive to time perturbation, and show no sensitivity to feature perturbation.

In the edge rewiring task, the quality of classical metrics vary significantly depending on the function descriptor and distance measure. For example, using activity rate with KS distance outperforms MMD, as KS is more sensitive to large deviations in cumulative distributions. In contrast, MMD smooths out these localized shifts, reducing sensitivity. The NC descriptor performs comparatively poorly across all measures, as the statistic often remains unchanged when edges are rewired.

In the time perturbation task, the JL-Metric outperforms all baseline metrics. Many traditional metrics assume i.i.d. relationships between function descriptors, limiting their ability to capture temporal changes unless these are indirectly reflected through topological shifts. The JL-Metric, by contrast, is more expressive, capturing both temporal and topological changes directly.

In the event perturbation task, the JL-Metric is the only metric sensitive to the perturbation. This is expected: Event permutation preserves topology, rendering classical metrics ineffective. Additionally, because the perturbation alters feature-topology relationships while keeping the features themselves unchanged, feature-based baselines fail to detect differences, as they are insensitive to feature order. In contrast, the JL-Metric captures dependencies between features and topology.

It is important to note that feature-based metrics are inherently insensitive to changes in topology, just as topological metrics are insensitive to changes in features. However, the JL-Metric shows robust sensitivity to both types of perturbation. Unlike traditional methods, our approach is agnostic to specific topological details and consistently performs well across all three types of perturbations.

## 4.2 MEASURING DIVERSITY

We assess each metric's ability to capture the diversity of generated data by focusing on the failure modes outlined in Section 2.4: *mode droppiong* and *mode collapse*.

To simulate these phenomena, we first train a Temporal Graph Network (TGN) (Rossi et al., 2020), a widely used discriminative model for CTDGs, on each dataset $\mathcal{G}_r$ (see Appendix D for training details). The trained TGN includes a *memory bank* that contains learned embeddings for each node. We apply affinity propagation (Frey & Dueck, 2007) to cluster these node embeddings into $k$ modes.

For *mode dropping*, we generate perturbed graph $\mathcal{G}_g(p)$ by removing modes with probability $p$ from $\mathcal{G}_r$ and replacing events $c(t_i)$ involving those modes with randomly selected events from the remaining modes. Specifically, for each event $c(t_i) = (\text{src}, \text{dst}, t_i, \mathbf{e}_{\text{src,dst}}(t_i))$ (Equation 1), if either the source node (src) or the destination node (dst) belongs to a mode that is being removed, the event is replaced with a randomly selected event from nodes in modes that are not dropped.

Table 1: Summary of metric performance across experiments. Values represent median ± standard error, with error bars omitted from first five columns in favor of distributional details in Figure 1.

| Metric | Perturbation Correlation (*Median Spearman Score*) | | | | | Sample Efficiency *(min. events)* | Comp. Efficiency *(s/100 events)* |
|---|---|---|---|---|---|---|---|
| | Edge Rewiring | Time Perturbation | Event Permutation | Mode Dropping | Mode Collapse | | |
| **Topological Metrics** | | | | | | | |
| Node Degree (KS) | 0.976 | 0.903 | — | 0.794 | 0.963 | $\mathbf{3 \pm 1}$ | $8.41 \pm 0.51$ |
| Node Degree (MMD) | **1.000** | 0.927 | — | 0.733 | **1.000** | $3 \pm 0$ | $8.56 \pm 0.35$ |
| LCC (KS) | 0.976 | 0.830 | — | 0.818 | 0.976 | $5 \pm 1$ | $11.99 \pm 0.68$ |
| LCC (MMD) | 0.952 | 0.842 | — | 0.806 | 0.770 | $4 \pm 0$ | $10.08 \pm 0.74$ |
| NC (KS) | 0.915 | 0.830 | — | 0.718 | 0.927 | $\mathbf{3 \pm 1}$ | $10.25 \pm 0.08$ |
| NC (MMD) | 0.988 | 0.685 | — | 0.552 | 0.976 | $\mathbf{3 \pm 1}$ | $10.77 \pm 0.15$ |
| PLE (KS) | 0.964 | 0.915 | — | 0.745 | 0.963 | $7 \pm 1$ | $9.68 \pm 0.10$ |
| PLE (MMD) | 0.976 | 0.830 | — | 0.685 | 0.976 | $8 \pm 1$ | $10.03 \pm 0.21$ |
| Activity Rate (KS) | 0.985 | — | — | 0.903 | **1.000** | $\mathbf{3 \pm 0}$ | $\mathbf{0.12 \pm 0.01}$ |
| Activity Rate (MMD) | 0.645 | — | — | 0.176 | 0.522 | $\mathbf{3 \pm 1}$ | $0.21 \pm 0.01$ |
| **Feature Metrics** | | | | | | | |
| Kullback-Leibler Div. | — | — | — | 0.103 | 0.649 | — | $0.72 \pm 0.02$ |
| Jensen-Shannon Div. | — | — | — | 0.685 | 0.842 | — | $0.69 \pm 0.04$ |
| KS | — | — | — | 0.488 | 0.713 | — | $0.54 \pm 0.04$ |
| MMD | — | — | — | 0.286 | 0.494 | — | $0.82 \pm 0.05$ |
| **JL-Metric (Ours)** | 0.976 | **0.944** | **0.988** | **0.915** | 0.988 | $\mathbf{3 \pm 1}$ | $1.05 \pm 0.09$ |

For *mode collapse*, we construct $\mathcal{G}_g(p)$ by replacing a randomly selected event $c(t_i)$ with the cluster centroid's event with probability $p$. In this process, nodes of $c(t_i)$ are rewired to those of the nearest mode centroid, and feature $\mathbf{e}_{\text{src,dst}}(t_i)$ is replaced by the mean feature of all events in that mode.

**Results:** Figure 1 (bottom two rows) visualizes the distributions of correlations across all datasets and random seeds, with median correlations shown in Table 1. In the mode dropping experiment, many classical metrics show suboptimal performance, with median correlations often below 0.80. However, certain metrics, such as activity rate (KS) and LCC, perform better, likely due to their ability to capture long-term temporal dependencies and global topological structure, respectively. These global perspectives are important for detecting subtle perturbations like mode dropping. The JL-Metric consistently achieves high correlations across both experiments.

## 4.3 Sample and Computational Efficiency

To evaluate sample efficiency, we follow the approach described in Section 2.4. We let $G'_r$ and $G''_r$ be subsets of each real-world dataset and $G'_g$ be a subset of the Grid dataset. The sample efficiency results are summarized in Table 1. Overall, all tested metrics demonstrate strong sample efficiency, consistent with results seen in the static graph domain (Thompson et al., 2022). However, topological metrics that rely on global features, such as LCC and PLE, exhibit comparatively lower sample efficiency. This outcome is intuitive, as these global structures typically require a larger number of events to manifest in the induced static graph.

The final property we examine is computational efficiency. To provide practical comparisons, we benchmark each metric's runtime across the entirety of each dataset, reporting results in seconds per 100 events, with hardware details in Appendix D. Results, summarized in Table 1, show that classical snapshot-based metrics are slower due to the need for explicit snapshot instantiation. In contrast, the activity rate and JL-Metric are much faster, as they avoid snapshot instantiation.

## 5 Conclusion

In this work, we introduced a novel approach for evaluating generative models of dynamic graphs by applying Johnson-Lindenstrauss transformations directly to dynamic graph data. Our method addresses key limitations of traditional metrics, providing an efficient, unified, and scalar measure that is sensitive to changes in both graph topology and features. Through comprehensive empirical evaluation, we demonstrated its effectiveness in capturing the essential properties of dynamic graphs.

REPRODUCIBILITY STATEMENT

We take several steps to ensure that our work is fully reproducible. The JL-metric algorithm is clearly defined in Section 3, with additional details in Appendix B. Baseline methods are fully described, with mathematical definitions provided in Appendix A. Experimental details, including hyperparameter search and the software and hardware used, are discussed in Appendix D. We use open source datasets with detailed descriptions in Appendix C. Additionally, the supplementary material includes an anonymized version of the code used for our experiments.

ACKNOWLEDGMENTS

This research used resources of the Argonne Leadership Computing Facility, a U.S. Department of Energy (DOE) Office of Science user facility at Argonne National Laboratory and is based on research supported by the U.S. DOE Office of Science-Advanced Scientific Computing Research Program, under Contract No. DE-AC02-06CH11357. Additional funding support comes from the National Science Foundation (CCF-2119184 CNS-2313190 CCF-1822949 CNS-1956180). RW gratefully acknowledges the support of NSF DMS-2023109, DOE DE-SC0022232, the NSF-Simons National Institute for Theory and Mathematics in Biology (NITMB) through NSF (DMS-2235451) and Simons Foundation (MP-TMPS-00005320), and the Margot and Tom Pritzker Foundation.

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

# A  ADDITIONAL DETAILS ON CLASSICAL DGGM METRICS

Here, we overview the statistics used to compute function descriptor $\mathbf{x}_i$ for the baseline generative models discussed in Section 2.2 of the main text: TagGen (Zhou et al., 2020), TIGGER (Gupta et al., 2022), Dymond (Zeno et al., 2021), and TG-GAN (Zhang et al., 2021). As discussed in Section 2.2, TagGen and TIGGER use only snapshot-based metrics while Dymond and TG-GAN use both snapshot-based and node behavior metrics. TagGen uses mean node degree, claw count, wedge count, power law exponent of degree distribution (PLE), largest connected component (LCC), and number of components (NC). TIGGER uses mean node degree, wedge count, triangle count, relative edge distribution entropy, global clustering coefficient, mean betweenness centrality, mean closeness centrality, PLE, LCC, and NC. Dymond uses the following snapshot-based statistics: density, average local clustering coefficient, s-metric, and LCC. Additionally, Dymond uses the following node behavior metrics: node activity rate, node temporal degree distribution, node clustering coefficient, node closeness centrality, node connected component size. TG-GAN uses the following snapshot-based metrics: mean node degree, average group size, average group number, and mean coordination number. TG-GAN additionally uses the following node behavior metrics: group size, average group size, mean coordination number, mean group number, mean group notation. It is important to that Dymond additionally uses a two-dimensional KS distance calculation on the first and third quartile of the node behavior statistics, in order to attempt to align the node distributions of the generated and ground-truth graph. Additional details are included in their work.

In our work, we choose representative baselines that are both used commonly and capture different attributes of the graph. We select average node degree as a way of capturing *local* topology. In contrast, we select LCC and NC to capture more global features of the graph. We select PLE in order to explicitly capture edge behavior of the graphs. We select the node behavior metric node activity rate due to ease of calculation and its demonstrated high sensitivity in prior work.

Next, we provide formal definitions for each of the baseline metrics used in our main work. These definitions are standard in the literature but are repeated here for convenience.

Let $\mathcal{G} = (V, E)$ be an undirected static graph, where $V$ is the set of nodes and $E$ is the set of edges. For a node $v \in V$, let $d(v)$ denote its degree. Let $F$ denote the set of all connected components in $\mathcal{G}$, where each component $f \in F$ is a set of connected nodes.

**Mean Degree:**  The mean node degree for all nodes in the graph is defined as:

$$\mu = \frac{1}{|V|} \sum_{v \in V} d(v).$$

**Largest Connected Component (LCC):**  The LCC represents the size of the largest connected component in the graph, defined as:

$$\max_{f \in F} \|f\|,$$

where $\|f\|$ represents the size of a connected component $f$.

**Number of Connected Components (NC):**  The number of connected components is simply $|F|$.

**Power-Law Exponent (PLE):**  The power-law exponent quantifies the scale of the power-law distribution in the degree sequence of the graph, computed as:

$$1 + |V| \left( \sum_{v \in V} \log \left( \frac{d(v)}{d_{\min}} \right) \right)^{-1},$$

where $d_{\min}$ is the minimum degree.

**Node Activity Rate:**  The node activity rate is defined as the number of events involving each node in the graph. Formally, for a given node $v \in V$, the activity rate $\alpha(v)$ is computed as:

$$\alpha(v) = \sum_{i=1}^{k} [\delta(v, \mathrm{src}_i) + \delta(v, \mathrm{dst}_i)],$$

where $k$ is the total number of events in the temporal graph, $\text{src}_i$ is the source node of event $i$, $\text{dst}_i$ is the destination node of event $i$, and $\delta(x, y)$ is the Kronecker delta function. The activity rate for all nodes can be represented as a 1D tensor where the $i$-th entry is the number of events involving node $i$.

# B ADDITIONAL DETAILS FOR JL-METRIC

## B.1 ADDITIONAL IMPLEMENTATION DETAILS

Here, we describe how normalization may be applied to the JL-Metric algorithm presented in Section 3 of the main work and provide more information regarding the structured random matrix (SRM) class used in our experiments.

In practice, we can normalize the event features and time stamps of the CTDG. Thus, the event representation $\tilde{c}(t_i)$ introduced in Section 3 and reproduced below for convenience:

$$\tilde{c}(t_i) = (t_i, \mathbf{e}_{\text{src,dst}}(t_i))$$

can be modified by normalization function $\zeta$:

$$\tilde{c}_\zeta(t_i) = (\zeta(t_i), \zeta(\mathbf{e}_{\text{src,dst}}(t_i))),$$

Here, the normalization function is applied separately to the time stamp and each feature channel in feature vector $\mathbf{e}_{\text{src,dst}}(t_i)$. Relevant statistics for the normalization can be collected from the original CTDG representation (Equation 1). For our experiments, we apply min-max normalization (Patro, 2015) and thus collect min and max values from the timestamp and each feature channel.

We find normalization improves performance as it aides balancing the importance of each feature and the timestamps, decoupling the magnitude of an attribute from its performance.

As discussed in Section 3, while orthogonal random matrices may be used for $W_1$ and $W_2$ in our JL-metric calculation, in practice we prefer to use a structured random matrix (SRM). We specifically experiment with using the SR-product matrices introduced by Choromanski et al. (2017). These matrices take the form $W = HD$, where $H \in \mathbb{R}^{l \times l}$ is a normalized Hadamard matrix and $D$ is a diagonal matrix with Rademacher random variables on the diagonal.

The normalized Hadamard matrix $H_l$ is defined recursively as:

$$H_1 = (1), \quad H_l = \frac{1}{\sqrt{2}} \begin{pmatrix} H_{l/2} & H_{l/2} \\ H_{l/2} & -H_{l/2} \end{pmatrix}$$

The diagonal matrix $D$ is populated with i.i.d. Rademacher random variables, i.e., $D_{ii} \sim \text{Unif}(\{-1, 1\})$. Note that this definition produces a square matrix. However, rows or columns can be removed as needed to achieve the desired dimensions. Together, the product of these matrices $W = HD$ forms the structured random matrix we use in place of fully random orthogonal matrices. Choromanski et al. (2017) demonstrates that such a matrix has performance similar to or better than orthogonal random baselines. We find empirically that for the JL-metric, choice of random matrix has a negligible effect on metric sensitivity. Thus, as described in our main work, we recommend the use of SRMs due their computational advantages.

## B.2 LIMITATIONS

Throughout our experiments, we identified potential limitations to our approach, which we summarize here.

While our metric is designed to be domain-agnostic, providing a unified assessment of dynamic graphs across application domains, it may not replace specialized metrics that are important for capturing specific graph properties in certain applications. For instance, in molecular and protein studies, metrics such as ring counts and other specialized structural features may be of particular

importance. In such cases, classical domain-specific metrics remain important for thorough analysis. Our method is intended to complement these specialized metrics by offering a general framework for evaluating dynamic graphs, rather than to replace them.

Our method relies on ordering nodes based on the timestamp of their first appearance to create a consistent representation of the dynamic graph. In rare situations where multiple nodes first appear at the exact same timestamp, this could introduce ordering ambiguity, similar to the graph isomorphism problem encountered in static graph analysis (Sato, 2020; Xu et al., 2019). However, CTDGs typically have continuous timestamps with high precision, making such instances uncommon. If nodes do share the same initial timestamp, secondary attributes such as node feature values could be used to establish a consistent ordering. Addressing this scenario is beyond the scope of our current work but represents an area for potential future exploration.

Our approach involves two hyperparameters related to the dimensions of the node and graph descriptors used in the random projections. Similar to prior random network-based methods (Thompson et al., 2022; Xu et al., 2018), there is no universally optimal choice for these hyperparameters. In our work, we perform a grid search to select reasonable values for our experiments as described in Appendix D.1. However, practitioners should be aware that the choice of descriptor dimensions can affect metric performance and ensure consistent dimensions when comparing metric performance. Future work could focus on developing automatic methods for hyperparameter selection.

## C  DATASET DETAILS

The Reddit, Wikipedia, LastFM, and MOOC datasets are bipartite temporal interaction graphs publicly released by Kumar et al., 2019.

In the Reddit dataset, the two node types are users and subreddits (communities within Reddit). An interaction (edge) is formed when a user engages with a subreddit, such as by posting. The Wikipedia dataset similarly contains users and pages as node types, with edges representing user edits to pages. For both datasets, interactions are timestamped, and event features are derived from text data.

The LastFM dataset captures user interactions with songs and thus has two node types: users and songs. Unlike the others, this dataset does not include specific feature attributes for interactions.

In the MOOC dataset, nodes represent users (students) taking a "massive open online course" and actions (interactions with course material). Edges form when a user interacts with a course, such as by clicking on videos or interactive content.

We make two major changes to the JODIE datasets: First, we shorten all datasets to the first 1,000 interactions, as calculating many classical graph metrics at the Nyquist rate is impractical otherwise. Second, we augment the LastFM dataset, which lacks features, by assigning a feature value of 1 to each event. This allows us to compute the JL-Metric (Section 3) without modifying the algorithm.

The Grid dataset, described in Section 4, is a grid-like CTDG where each node connects to its neighbors at regular time intervals $t = \{t_1, ..., t_n\}$. The event features $\mathbf{e}_{\text{src, dst}}(t_i) = h(\text{src}, \text{dst}, t_i)$ introduce explicit dependencies on both the temporal and topological aspects of the graph. Specifically, for our experiments, we define $h(\text{src}, \text{dst}, t_i)$ to produce two features based on the temporal and node information: $h(\text{src}, \text{dst}, t_i) = (\text{src} \cdot t_i, \text{dst} + t_i)$.

Table 2 contains statistics for each dataset regarding these datasets after our preprocessing described above. Additionally, our accompanying code contains a tool that provides detailed dataset statistics for a given interaction count. Finally, we refer interested readers to Kumar et al., 2019 for additional dataset details.

## D  EXPERIMENTAL DETAILS

Here, we detail our hyperparameter search for the JL-Metric, training and hyperparameters for the Temporal Graph Network (TGN) (Rossi et al., 2020) used in the diversity experiments (Section 4.2), and provide information on the hardware and software used.

| Dataset | # Nodes | # Interactions | Snapshots (Nyquist) | $|\mathbf{e}_{src,dst}|$ |
|---------|---------|----------------|---------------------|-------------|
| Reddit | 852 | 1,000 | 4,887 | 172 |
| Wikipedia | 377 | 1,000 | 27,365 | 172 |
| MOOC | 147 | 1,000 | 71,233 | 4 |
| LastFM | 385 | 1,000 | 848,680 | 1 |
| Grid | 539 | 1,000 | 9,000 | 2 |

Table 2: Statistics of Dynamic Graph Datasets

### D.1 JL-METRIC HYPERPARAMETER SEARCH

As discussed in Section 3, the JL-metric has two tunable hyperparameters: node event embedding size $n$ and descriptor size $o$. In our experiments, we conduct a progressive search for reasonable values of $n$ and $o$ by gradually increasing both until performance gains begin to stagnate ($< 1\%$ change). Starting with $n, o = 25$, we incrementally test values up to 200 in steps of 25. For each pair, we monitor the median Spearman score across all perturbation experiments (Sections 4.1 and 4.2). Once the performance improvements plateau—indicating diminishing returns with higher values—we select the smallest $n$ and $o$ that yield near-optimal results. In our case, we select $n = 100$ and $o = 100$. As anticipated from the Johnson-Lindenstrauss lemma (Section 3), we expect the required $n$ to grow with the number of nodes $z$, and required $o$ to grow with the number of graphs being compared. Future work should empirically verify these scaling relationships.

### D.2 TEMPORAL GRAPH NETWORK TRAINING DETAILS

As detailed in the main work, in order to cluster nodes into *modes* for the diversity experiments in Section 4.2, we train a Temporal graph network (TGN) (Rossi et al., 2020). To do so, we train on the supervised task of *link prediction*, as detailed in the original work. We additionally keep all training details the same: We use the Adam optimizer, binary cross entropy loss, and a $70\% - 15\% - 15\%$ chronological train-validation-test split. Given that Rossi et al. (2020) experiments with the same datasets we use (Jodie (Kumar et al., 2019)), we forgo hyperparameter search and use the reported values from that work. These are repeated here for convenience: memory dimension $= 172$, node embedding dimension $= 100$, time embedding dimension $= 100$, number of attention heads $= 2$, and dropout $= 0.1$. We select the model with the best validation loss.

### D.3 SOFTWARE AND HARDWARE TOOLS

We primarily rely on the Pytorch geometric (Fey & Lenssen, 2019) and NetworkX (Hagberg et al., 2008) open-source Python libraries for static graph representations. Given its importance to runtime benchmarking and overall reproducability, we provide a full list of software libraries used in our experiments, as well as their respective versions, in the Supplementary material. We additionally provide an anonymized version of our code in the Supplementary material. All metrics are tested on a AMD EPYC 7713 64-Core Processor and the Red Hat Enterprise Linux 9.3 operating system.

## E DATASET SPECIFIC RESULTS

The results presented in Figure 1 and Table 1 in the main work are aggregated across all tested datasets. Here, we include results of the sensitivity analysis (measuring fidelity and diversity) for individual datasets. Tables 3—7 present the results for the Grid, Reddit, Wikipedia, MOOC, and LastFM datasets, respectively. As noted in the main work, the LastFM dataset does not contain features and thus the *Event Permutation* values are left as N/A in Table 7.

Table 3: Perturbation Correlation results for the *Grid* dataset.

| Metric | Perturbation Correlation (*Median Spearman Score*) | | | | |
|---|---|---|---|---|---|
| | Edge Rewiring | Event Permutation | Time Perturbation | Mode Collapse | Mode Dropping |
| **Grid Dataset** | | | | | |
| Node Degree (KS) | 0.988 | — | 0.988 | 0.879 | 0.755 |
| Node Degree (MMD) | 0.988 | — | 0.988 | 1.000 | 0.782 |
| LCC (KS) | 0.988 | — | 0.903 | 1.000 | 0.915 |
| LCC (MMD) | 0.867 | — | 0.927 | 1.000 | 0.952 |
| NC (KS) | 0.927 | — | 0.877 | 0.514 | 0.623 |
| NC (MMD) | 0.976 | — | 0.886 | 0.830 | 0.527 |
| PLE (KS) | 0.976 | — | 0.988 | 0.891 | 0.298 |
| PLE (MMD) | 0.976 | — | 0.976 | 1.000 | 0.879 |
| Activity Rate (KS) | 0.988 | — | — | 1.000 | 0.297 |
| Activity Rate (MMD) | 0.603 | — | — | 0.522 | 0.378 |
| **Feature Metrics** | | | | | |
| Kullback-Leibler Div. | — | — | — | — | — |
| Jensen-Shannon Div. | — | — | — | 0.796 | 0.522 |
| KS | — | — | — | 0.960 | 0.123 |
| MMD | — | — | — | — | — |
| **JL-Metric (Ours)** | 1.000 | 0.988 | 0.979 | 1.000 | 0.503 |

Table 4: Perturbation Correlation results for the *Reddit* dataset.

| Metric | Perturbation Correlation (*Median Spearman Score*) | | | | |
|---|---|---|---|---|---|
| | Edge Rewiring | Event Permutation | Time Perturbation | Mode Collapse | Mode Dropping |
| **Grid Dataset** | | | | | |
| Node Degree (KS) | 0.988 | — | 0.527 | 0.964 | 0.855 |
| Node Degree (MMD) | 1.000 | — | 0.491 | 0.976 | 0.806 |
| LCC (KS) | 0.988 | — | 0.729 | 0.231 | 0.815 |
| LCC (MMD) | 0.988 | — | 0.782 | — | 0.782 |
| NC (KS) | 0.891 | — | 0.370 | 0.985 | 0.988 |
| NC (MMD) | 0.988 | — | 0.455 | 0.988 | 0.976 |
| PLE (KS) | 0.988 | — | 0.539 | 0.963 | 0.745 |
| PLE (MMD) | 0.758 | — | 0.539 | 0.976 | 0.248 |
| Activity Rate (KS) | 0.952 | — | — | 1.000 | 1.000 |
| Activity Rate (MMD) | 0.748 | — | — | 0.498 | 0.097 |
| **Feature Metrics** | | | | | |
| Kullback-Leibler Div. | — | — | — | 0.915 | 0.782 |
| Jensen-Shannon Div. | — | — | — | 0.927 | 0.903 |
| KS | — | — | — | 0.742 | 0.488 |
| MMD | — | — | — | 0.781 | 0.535 |
| **JL-Metric (Ours)** | 0.903 | 1.000 | 0.960 | 0.976 | 0.988 |

Table 5: Perturbation Correlation results for the *Wikipedia* dataset.

| Metric | Perturbation Correlation (*Median Spearman Score*) | | | | |
|---|---|---|---|---|---|
| | Edge Rewiring | Event Permutation | Time Perturbation | Mode Collapse | Mode Dropping |
| **Grid Dataset** | | | | | |
| Node Degree (KS) | 0.924 | — | 0.830 | 0.902 | 0.648 |
| Node Degree (MMD) | 1.000 | — | 0.782 | 0.988 | 0.491 |
| LCC (KS) | 0.939 | — | 0.794 | 0.172 | 0.782 |
| LCC (MMD) | 0.794 | — | 0.733 | — | 0.733 |
| NC (KS) | 0.952 | — | 0.717 | 0.988 | 0.939 |
| NC (MMD) | 0.988 | — | 0.600 | 0.988 | 0.830 |
| PLE (KS) | 0.867 | — | 0.879 | 0.957 | 0.806 |
| PLE (MMD) | 1.000 | — | 0.806 | 0.988 | 0.661 |
| Activity Rate (KS) | 0.976 | — | — | 1.000 | 0.988 |
| Activity Rate (MMD) | 0.733 | — | — | 0.793 | — |
| **Feature Metrics** | | | | | |
| Kullback-Leibler Div. | — | — | — | 0.830 | 0.273 |
| Jensen-Shannon Div. | — | — | — | 0.879 | 0.879 |
| KS | — | — | — | 0.778 | 0.711 |
| MMD | — | — | — | 0.637 | 0.661 |
| **JL-Metric (Ours)** | 0.867 | 0.988 | 0.944 | 0.952 | 0.952 |

Table 6: Perturbation Correlation results for the *MOOC* dataset.

| Metric | Perturbation Correlation *(Median Spearman Score)* | | | | |
|---|---|---|---|---|---|
| | Edge Rewiring | Event Permutation | Time Perturbation | Mode Collapse | Mode Dropping |
| **Grid Dataset** | | | | | |
| Node Degree (KS) | 0.939 | — | 0.891 | 0.964 | 0.721 |
| Node Degree (MMD) | 0.867 | — | 0.939 | 1.000 | 0.685 |
| LCC (KS) | 0.720 | — | 0.709 | 0.976 | 0.815 |
| LCC (MMD) | 0.964 | — | 0.745 | 0.770 | 0.600 |
| NC (KS) | 0.333 | — | 0.719 | 0.921 | 0.588 |
| NC (MMD) | 0.418 | — | 0.603 | 0.906 | 0.406 |
| PLE (KS) | 0.879 | — | 0.842 | 0.842 | 0.673 |
| PLE (MMD) | 0.842 | — | 0.782 | 0.915 | 0.648 |
| Activity Rate (KS) | 0.967 | — | — | 1.000 | 0.624 |
| Activity Rate (MMD) | 0.348 | — | — | 0.522 | 0.350 |
| **Feature Metrics** | | | | | |
| Kullback-Leibler Div. | — | — | — | 0.648 | 0.697 |
| Jensen-Shannon Div. | — | — | — | 0.830 | 0.806 |
| KS | — | — | — | 0.569 | 0.731 |
| MMD | — | — | — | 0.541 | 0.695 |
| **JL-Metric (Ours)** | 0.952 | 0.988 | 0.527 | 0.964 | 0.661 |

Table 7: Perturbation Correlation results for the *LastFM* dataset.

| Metric | Perturbation Correlation *(Median Spearman Score)* | | | | |
|---|---|---|---|---|---|
| | Edge Rewiring | Event Permutation | Time Perturbation | Mode Collapse | Mode Dropping |
| **Grid Dataset** | | | | | |
| Node Degree (KS) | 0.988 | — | 0.964 | 1.000 | 0.888 |
| Node Degree (MMD) | 1.000 | — | 0.939 | 1.000 | 0.818 |
| LCC (KS) | 1.000 | — | 0.976 | 1.000 | 0.952 |
| LCC (MMD) | 0.988 | — | 0.964 | 1.000 | 0.939 |
| NC (KS) | 0.952 | — | 0.891 | 0.915 | 0.612 |
| NC (MMD) | 0.939 | — | 0.891 | 0.939 | 0.491 |
| PLE (KS) | 0.988 | — | 0.927 | 0.988 | 0.842 |
| PLE (MMD) | 1.000 | — | 0.867 | 0.964 | 0.903 |
| Activity Rate (KS) | 1.000 | — | — | 1.000 | 0.952 |
| Activity Rate (MMD) | 0.603 | — | — | 0.711 | — |
| **Feature Metrics** | | | | | |
| Kullback-Leibler Div. | — | — | — | — | — |
| Jensen-Shannon Div. | — | — | — | — | — |
| KS | — | — | — | — | — |
| MMD | — | — | — | — | — |
| **JL Embedding (Ours)** | 1.000 | N/A | 0.985 | 0.988 | 0.927 |

