# OpenReview forum: "Quality Measures for Dynamic Graph Generative Models"
_ICLR.cc/2025/Conference — ICLR 2025 Spotlight_

### Official Review · Reviewer_f4HN · 2024-10-28

**Soundness:** 3
**Presentation:** 3
**Contribution:** 3
**Rating:** 8
**Confidence:** 2

**Summary:**

The paper presents a novel metric designed for evaluating generative models of dynamic graphs, where both topology and features evolve over time. The authors propose a new metric based on the Johnson-Lindenstrauss (JL) lemma, which leverages random projections to create an expressive, scalar measure that captures the complex dependencies in dynamic graphs, overcoming limitations in current evaluation methods.

Current metrics for evaluating dynamic graph generative models (DGGMs) rely on static snapshots, and therefore lose the temporal dependencies. Moreover, current metrics fail to capture node and edge features and their relation to the graph topology. They are also only sensible to specific properties resulting to the need of multiple metrics. Many of these metrics are also computationaly inefficient.

To address these limitations, the authors propose a new Johnson-Lindenstrauss-based (JL) metric, inspired by work in the static graph domain and image-based evaluations. The metric applies random projections directly to continuous-time dynamic graph data, effectively embedding the variable-length sequence of graph events into a fixed-dimensional vector space. This transformation preserves the similarity of data across temporal interactions and node features while avoiding the computational cost of explicit snapshot instantiation.

The author justify the use of random projections on the Johnson-Lindenstrauss lemma, which asserts that random orthogonal projections can approximately preserve the distance between data points. This property allows the proposed metric to map dynamic graph events of varying lengths into a unique dimension.

Experiments are conducted on both real-world datasets (e.g., Reddit, Wikipedia, LastFM) and synthetic datasets. They show that the JL metric provides consistent, high-fidelity measurements across topological and temporal changes, with reduced computational overhead.

**Strengths:**

The paper is well written and structured, making it easy for readers to follow.

By leveraging the Johnson-Lindenstrauss lemma for random projections, this method offers several advantages, including the ability to capture temporal dependencies, unify topology and feature changes into a single scalar metric, and reduce computational cost.

The empirical evaluation demonstrates the effectiveness of the new metric. The experiments validate the interest of the method and its practical utility.

Additionally, Section 3 provides new theoretical insights into why random-network-based metrics may be effective in general, and for dynamic graphs in particular.

**Weaknesses:**

The methodological novelty of the proposed approach is somewhat limited, as similar frameworks have already been applied, including to static graphs. The authors themselves acknowledge this by stating that they "follow recent analogous work in the static graph domain by Thompson et al., 2022." The contribution is therefore limited.

The applicability of the proposed metric is focused on continuous-time dynamic graph generative models (CTDGs) with a given initial graph. It is a relatively small field within dynamic graph research, where most studies adopt a supervised learning setting. Moreover, new metrics for CTDGs can be integrated within papers introducing novel generative models, as it have been the case for instance in Zhang et al. (2021). The potential impact of this work may be limited.

The paper does not include a discussion of the limitations of the method. For instance, it does not address the fact that the metric evaluates only the changes in the graph over time rather than the graph structure itself, limiting the possible application of the metric. Scalability could be an issue, for example, when applying the method to large graphs. These are just examples and a paragraph on some limitations of the method would be insight full.

The paper does not provide practical recommendations for applying the metric to common datasets. Specifically, there is no guidance on selecting the optimal number of samples, events, or the dimensions of descriptors, which could help in effectively using the metric on various datasets.

Minor comment:
I think that there is a small typo in the formula at the end of line 119.

**Questions:**

Please, could you comment on the limitations mentioned above? On the fact that the metric only evaluates changes rather than the graph distribution itself and on the scalability issue.

Could you also comment on the small number of dynamic graph generative models?

---

> ### Author Response · Authors · 2024-11-20
>
> Thanks for your thorough review and for highlighting the strengths of our work, including the clarity of our presentation, the advantages of leveraging the JL lemma, and the effectiveness demonstrated by our empirical evaluations. We appreciate your constructive feedback and address your concerns individually below:
>
> > The methodological novelty of the proposed approach is somewhat limited, as similar frameworks have already been applied, including to static graphs. The authors themselves acknowledge this by stating that they "follow recent analogous work in the static graph domain by Thompson et al., 2022." The contribution is therefore limited.
>
> We respectfully disagree with the assessment that the novelty of our approach is limited. To summarize, while our work is indeed motivated by random network-based metrics (as described in Section 2.3), our work significantly extends it into the dynamic graph domain with novelties that address complexities specific to dynamic evolution. Additionally, prior works did not actually use the JL lemma directly; we believe we are the first to do so and to establish a tentative link between prior work on random network-based metrics and the JL lemma. We believe our contributions significantly extend beyond prior static graph methods in several key ways which we highlight in detail below:
>
> 1. We discuss and address several limitations of current static-based metric approaches that are problematic specifically for dynamic graphs. For example, faulty i.i.d. assumptions (lines 45–49; 162–164; 316–322) and reliance on static snapshot construction (lines 144–150) are issues not present in static graphs but critical in dynamic graphs. One reason why the JL-metric overcomes these limitations by effectively modeling temporal dependencies—an aspect unique to dynamic graphs.
>
> 2. In Section 4, we conduct the first empirical evaluation of existing dynamic graph metrics. We design sensitivity analyses specifically to assess metrics for evolving graphs. When justifying our design choices, we emphasize non-uniform evolution and temporal dependencies (e.g., lines 457–460; 463–465; 513–516), which are unique challenges in dynamic graphs.
>
> 3. Unlike many classical methods that rely on constructing explicit adjacency matrices, our approach operates directly on continuous-time data. This allows us to capture temporal dynamics while bypassing the tradeoff between information loss and runtime/memory demand inherent in discretization (lines 348-351). Such a tradeoff does not exist for static graphs.
>
> 4. Works such as [1] use randomly initialized neural networks as function descriptors without providing theoretical justification for their effectiveness. In contrast, in our work, we establish a tentative connection between these approaches and the JL lemma (Section 3). This insight allows us to develop our metric independently of existing neural networks for dynamic graphs, most of which still rely on static snapshots.

---

> ### Author Response · Authors · 2024-11-20
>
> > The applicability of the proposed metric is focused on continuous-time dynamic graph generative models (CTDGs) with a given initial graph. It is a relatively small field within dynamic graph research, where most studies adopt a supervised learning setting. Moreover, new metrics for CTDGs can be integrated within papers introducing novel generative models, as it have been the case for instance in Zhang et al. (2021). The potential impact of this work may be limited.
>
>
> We respectfully disagree with the notion that the applicability of our proposed metric is too niche. CTDGs are a general representation of dynamic graphs and thus our method can be used to compare other dynamic graphs via conversion to CTDG. Also, while our evaluation focuses on comparing dynamic graphs in the context of DGGMs, our method is general enough to assess dynamic graph similarity in _any context_. More details below:
>
> 1. CTDGs are a general representation of dynamic graphs (lines 100-105) which moreover provide a compact and flexible representation for modeling dynamic interactions (lines 116–117). Thus, the literature has increasingly focused on learning directly on CTDGs [2], reflecting their growing importance. Also, since other dynamic graphs can be converted to CTDGs, our metric is general and applicable to various types of dynamic graphs.
>
> 2. While our evaluation focuses on DGGMs, our method is able to assess dynamic graph similarity more generally. Nonetheless, we believe that DGGMs represent a significant and expanding area of research, as highlighted by the works we discuss [3-6] and other recent literature [7-13]. A recent survey [14], particularly Section 3.4, discusses the nascent yet rapidly developing nature of DGGMs. Additionally, a recent empirical study [15] highlights increasing interest in this area. Finally, a survey on generative models for static graphs [16] also emphasizes the importance of DGGMs as a future direction. This demonstrates that DGGMs are not a small niche but a significant and growing domain within graph research.
>
> 3. DGGMs have important real-world applications. As we discuss in lines 34–37, they are crucial for tasks such as modeling social network dynamics, biological systems, and communication networks. The DGGM papers cited above explore additional diverse applications, including protein folding, online shopping recommendation systems, and traffic simulation.
>
> 4. Finally, we believe that developing a properly expressive, domain-agnostic, and scalar metric, as proposed in our work, will provide researchers easier comparison and evaluation of DGGMs which can in turn further innovation in the field.
>
> We believe our metric is broadly applicable and addresses a significant need within the expanding field of DGGMs. We hope this clarifies the potential impact and relevance of our work.
>
> > The paper does not include a discussion of the limitations of the method. For instance, it does not address the fact that the metric evaluates only the changes in the graph over time rather than the graph structure itself, limiting the possible application of the metric. Scalability could be an issue, for example, when applying the method to large graphs. These are just examples and a paragraph on some limitations of the method would be insight full.
>
> Could you please clarify what you mean by "the metric evaluates only the changes in the graph over time rather than the graph structure itself"? Our proposed metric is indeed sensitive to changes in topology, even when temporal aspects remain unchanged. To illustrate this, please refer to our edge rewiring experiment in Section 4.1 (Figure 1, top row; lines 441–442 and 452–456). Here, we alter the graph's topology while keeping timestamps constant, and our metric demonstrates high sensitivity to these structural changes.
>
> We appreciate the importance of addressing limitations and have added a discussion to the manuscript (Appendix D.4). Below, we summarize key limitations:
>
> 1. _Domain-specific metrics:_ Our metric is domain-agnostic and may not replace specialized metrics crucial for specific graph properties (e.g., ring counts in molecular studies). These classical metrics remain essential for domain-specific needs, while our method provides a unified, general assessment.
>
> 2. _Ordering ambiguity:_ Our method orders nodes based on their first appearance timestamp. In rare cases where nodes share the same timestamp, ambiguity could arise. However, such scenarios are infrequent in high-resolution continuous-time graphs. See our response to Reviewer eyCj for more details.
>
> 3. _Hyperparameter sensitivity:_ Our approach involves two hyperparameters for descriptor dimensions. As with prior random network-based methods [1, 17], no universally optimal values exist. We use grid search to find reasonable values and provide insights in Appendix D.1. However, users should ensure consistent descriptor dimensions, as these can impact metric performance.

---

> > ### Author Response · Authors · 2024-11-20
> >
> > > The paper does not provide practical recommendations for applying the metric to common datasets. Specifically, there is no guidance on selecting the optimal number of samples, events, or the dimensions of descriptors, which could help in effectively using the metric on various datasets.
> >
> > Regarding the optimal number of samples (events), as with any other sample-based metric, using more samples improves the estimation of the underlying distribution and therefore the quality of the metric. A common validation method for DGGMs is to generate a synthetic graph with the same number of interactions as the ground truth graph (e.g., [4]). Our metric is flexible and can be applied in this setup as well as other commonly used configurations.
> >
> > For selecting the optimal dimensions of the descriptors, we perform a grid search to choose the appropriate parameters, progressively increasing the dimension of the descriptors until performance (median Spearman correlation across all experiments) stagnates (Appendix D.1 for details). Our analysis led to insights that may benefit practitioners using our method, which we have summarized in Appendix D.1. If you believe additional guidance is needed, we would be happy to expand on this section.
> >
> > > Minor comment: I think that there is a small typo in the formula at the end of line 119.
> >
> > Thanks for bringing this to our attention. We have fixed the typo in the newly uploaded document.
> >
> > > Please, could you comment on the limitations mentioned above? On the fact that the metric only evaluates changes rather than the graph distribution itself and on the scalability issue.
> >  Could you also comment on the small number of dynamic graph generative models?
> >
> >
> > Please see our responses above, including the highlighted recent DGGMs [3-13].
> >
> > Regarding scalability specifically, we would like to emphasize the memory and runtime complexities are linear and log-linear, respectively (lines 331-333).  Practically speaking, by avoiding explicit instantiation of static snapshots, our work leads to runtimes about one order of magnitude faster than discrete metrics, as found in our runtime benchmarking (Section 4.4; Table 1).
> >
> > We hope that we have sufficiently addressed your concerns.

---

> ### Author Response · Authors · 2024-11-20
>
> [1] Thompson, Rylee, et al. "On Evaluation Metrics for Graph Generative Models." International Conference on Learning Representations (ICLR), 2022.
>
> [2]  Kazemi, Seyed Mehran. "Dynamic graph neural networks." Graph Neural Networks: Foundations, Frontiers, and Applications (2022): 323-349.
>
> [3] Zhou, Dawei, et al. "A data-driven graph generative model for temporal interaction networks." Proceedings of the 26th ACM SIGKDD International Conference on Knowledge Discovery & Data Mining. 2020.
>
> [4] Gupta, Shubham, et al. "Tigger: Scalable generative modelling for temporal interaction graphs." Proceedings of the AAAI Conference on Artificial Intelligence. Vol. 36. No. 6. 2022.
>
> [5] Zeno, Giselle, Timothy La Fond, and Jennifer Neville. "Dymond: Dynamic motif-nodes network generative model." Proceedings of the Web Conference 2021. 2021.
>
> [6] Zhang, Liming, et al. "TG-GAN: Continuous-time temporal graph deep generative models with time-validity constraints." Proceedings of the Web Conference 2021. 2021.
>
> [7] Liu, Penghang, and Ahmet Erdem Sariyüce. "Using motif transitions for temporal graph generation." Proceedings of the 29th ACM SIGKDD Conference on Knowledge Discovery and Data Mining. 2023.
>
> [8] Clarkson, Jase, et al. "DAMNETS: A deep autoregressive model for generating Markovian network time series." Learning on Graphs Conference. PMLR, 2022.
>
> [9] Du, Yuanqi, et al. "Disentangled spatiotemporal graph generative models." Proceedings of the AAAI Conference on Artificial Intelligence. Vol. 36. No. 6. 2022.
>
> [10] Zhang, Wenbin, et al. "Disentangled dynamic graph deep generation." Proceedings of the 2021 SIAM International Conference on Data Mining (SDM). Society for Industrial and Applied Mathematics, 2021.
>
> [11] Zhang, Liming. "STGGAN: Spatial-temporal graph generation." Proceedings of the 27th ACM SIGSPATIAL International Conference on Advances in Geographic Information Systems. 2019.
>
> [12] Limnios, Stratis, et al. "Random Walk based Conditional Generative Model for Temporal Networks with Attributes." NeurIPS 2022 Workshop on Synthetic Data for Empowering ML Research. 2022.
>
> [13] Yousuf, Muhammad Irfan, and Suhyun Kim. "A generative model for time evolving networks." Knowledge and Information Systems 63.9 (2021): 2347-2363.
>
> [14] Gupta, Shubham, and Srikanta Bedathur. "A survey on temporal graph representation learning and generative modeling." arXiv preprint arXiv:2208.12126 (2022).
>
> [15] Souid, Houssem Eddine, et al. "Temporal Graph Generative Models: An empirical study." Proceedings of the 4th Workshop on Machine Learning and Systems. 2024.
>
> [16] Guo, Xiaojie, and Liang Zhao. "A systematic survey on deep generative models for graph generation." IEEE Transactions on Pattern Analysis and Machine Intelligence 45.5 (2022): 5370-5390.
>
> [17] Qiantong Xu, Gao Huang, Yang Yuan, Chuan Guo, Yu Sun, Felix Wu, and Kilian Weinberger. “An empirical study on evaluation metrics of generative adversarial networks.” arXiv preprint arXiv:1806.07755, 2018.

---

> > ### Comment · Reviewer_f4HN · 2024-11-28
> >
> > Thank for your comprehensive answer.
> >
> > I acknowedge to have underestimated this growing field of research. I also sensible to your arguments on novelty.
> > I revised my evaluations accordingly.

---

### Official Review · Reviewer_eyCj · 2024-11-01

**Soundness:** 3
**Presentation:** 3
**Contribution:** 3
**Rating:** 6
**Confidence:** 3

**Summary:**

The paper proposes a new metric for measuring similarities between temporal graphs, utilizing the input dimension agnostic property of random projection certified by the JL lemma. The metric is based on a node interaction history representation of a temporal graph, computed via first projecting individual node histories, followed by another random projection that fuses nodes. Experimental results demonstrate that the proposed metric achieves better fidelity and diversity than classic metrics, while being computationally efficient and sample efficient.

**Strengths:**

- Defining a suitable metric for assessing generation quality of temporal graphs is an important problem in graph generative modeling. The proposed metric is a novel approach that goes beyond the traditional way of using statistical summaries as quality measures.
- The proposed JL metric is shown to behave well empirically, especially in the event permutation sensitivity analysis.

**Weaknesses:**

- The proposed JL metric is stated to accommodate both topological information and feature information. While the overall assessments using sensitivity analysis have shown that JL indeed performs better than baselines, it would be more intuitive if the authors provide concrete evidences illustrating the sensitivity to some topological structures that exists in the evaluation datasets.
- In line 277 the authors proposed to use a simplified version of node history as node level presentation. The simplification essentially drops (some) interaction information, i.e., the interaction nodes' identity information. According to my understanding, this simplification inevitably looses capability to account for topological information.

**Questions:**

- As the authors use JL as their motivation for representation construction, I think it would be interesting if the authors provide the exact JL bounds that incurred during empirical evaluations: How well does JL compresses real world temporal graphs, according to the standard JL bound?

---

> ### Author Response · Authors · 2024-11-20
>
> Thank you for your thoughtful review and for recognizing the importance of our work with respect to assessing the generation quality of temporal graphs. We appreciate your constructive feedback and address your concerns and questions individually below.
>
> > The proposed JL metric is stated to accommodate both topological information and feature information. While the overall assessments using sensitivity analysis have shown that JL indeed performs better than baselines, it would be more intuitive if the authors provide concrete evidences illustrating the sensitivity to some topological structures that exists in the evaluation datasets.
>
> We agree that providing concrete examples illustrating the sensitivity of our JL-Metric to specific topological structures would enhance the intuition behind our results.
>
> One illustrative example we encountered involves the triangle count metric, a classical topology-based measure that uses the number of triangles in a graph as the descriptor. In our preliminary experiments, we found that this metric exhibited no sensitivity (Spearman correlation = 0‡) to the perturbations in the fidelity experiments (edge rewiring, event perturbation, and time permutation; Section 4.1) on the Reddit and Wikipedia datasets. This is unsurprising because both datasets are bipartite and therefore triangle-free by nature. Consequently, any topological changes that do not introduce triangles (such as in our fidelity experiments) go undetected by this metric.
>
> In contrast, our JL-Metric is sensitive to such changes because it does not rely on specific graph features like triangle counts. For instance, our JL-Metric shows a median Spearman correlation of 0.952 across the same experiments and datasets (minimum 0.87, maximum 1.00; see Appendix E for detailed results). While we excluded the triangle count metric from our final experiments due to this fundamental limitation, this example illustrates how traditional metrics can have blind spots that our approach addresses.
>
> ‡ Pedantically speaking, the Spearman correlation is undefined here, as there is no variation in the triangle metric.
>
> > In line 277 the authors proposed to use a simplified version of node history as node level presentation. The simplification essentially drops (some) interaction information, i.e., the interaction nodes' identity information. According to my understanding, this simplification inevitably looses capability to account for topological information.
>
> The representation $\mathcal{V}$ (line 269) does not lead to a loss of expressiveness in accounting for topological information. In fact, even if in each $\mathbf{v}_j \in \mathcal{V}$ the src and dst IDs are missing, it is still possible to recover the nodes that participated in an interaction occurred at a generic time $t_i$ because there are only two node representations, $\mathbf{v}_j$ and $\mathbf{v}_k$, that contain a reduced interaction representation $\tilde{c}(t_i)$ (lines 272 and 277) having the combination of timestamp and features that uniquely identify the interaction at time $t_i$.
>
> An interesting potential limitation to our work can occur where multiple nodes appear at the exact same timestamp, potentially introducing ordering ambiguity similar to the graph isomorphism problem in learning on static graphs (See [1, 2]). However, CTDGs typically have continuous timestamps with high precision, making such instances rare. In practice, if nodes do share the same initial timestamp, secondary attributes (e.g., feature values) can be used to establish a consistent ordering, though this is out of the scope of our work.
>
> > As the authors use JL as their motivation for representation construction, I think it would be interesting if the authors provide the exact JL bounds that incurred during empirical evaluations: How well does JL compresses real world temporal graphs, according to the standard JL bound?
>
> It is challenging to directly assess how our JL-Metric's practical performance compares to the theoretical JL bounds because the similarity between CTDGs is not precisely defined in a way that allows for straightforward quantification.
>
> Therefore, we rely on the empirical results presented in Section 4, which indicate that our JL-Metric effectively captures essential structural and temporal characteristics of real-world temporal graphs. Specifically, the metric demonstrates high sensitivity to various perturbations and failure modes of generative models, suggesting that it compresses the graphs sufficiently to preserve meaningful similarities and differences.
>
>
> [1] Sato, Ryoma. "A survey on the expressive power of graph neural networks." arXiv preprint arXiv:2003.04078 (2020).
>
> [2] Xu, Keyulu, et al. "How powerful are graph neural networks?" International Conference on Learning Representations (ICLR), 2019.

---

> > ### Comment · Reviewer_eyCj · 2024-11-22
> >
> > Thanks for your feedbacks. I will keep my score.

---

### Official Review · Reviewer_zdbk · 2024-11-04

**Soundness:** 3
**Presentation:** 3
**Contribution:** 3
**Rating:** 8
**Confidence:** 1

**Summary:**

The paper introduces a novel quality metric, JL-Metric, for evaluating generative models of dynamic graphs, addressing limitations in current metrics that treat temporal events as independent and fail to capture the integrated evolution of both graph topology and features. By leveraging the Johnson-Lindenstrauss lemma, the authors propose a method that uses random projections to measure similarity between dynamic graphs, resulting in an expressive, scalar metric applicable to continuous-time dynamic graphs. Empirical results suggest this metric achieves high fidelity and computational efficiency compared to traditional metrics.

**Strengths:**

S1: Unified Metric for Temporal Dynamics: The proposed metric overcomes the limitations of traditional methods by capturing dependencies between events and integrating both topological and feature dynamics, specifically for CTDG, which looks rational to me.

S2: High Efficiency and Practicality: The method’s use of random projections reduces runtime and memory demands, making it feasible for large-scale graph evaluation tasks, along with extensive evaluations, which looks comprehensive to me.

**Weaknesses:**

See questions.

**Questions:**

I am not highly specialized in this dynamic graph metric research area, but I do have a few general questions:


Q1: How does the method ensure robust sensitivity to subtle changes in node and edge features, especially when applied to simpler dynamic graphs or those with more complex interactions beyond temporal events? In theory, how do graph scale and interaction complexity influence the performance of this metric?

Q2: This paper covers a range of metrics and theoretical concepts. As a minor suggestion, it might enhance clarity to include a high-level figure illustrating the differences between the proposed metric and others, beyond just presenting post-experimental results.

---

> ### Author Response · Authors · 2024-11-20
>
> Thanks for your positive review of our paper and for your thoughtful comments. We're pleased that you recognize the value of our proposed metric in capturing temporal dynamics and its efficiency for large-scale graph evaluations. We're also glad you found our evaluations comprehensive. We address your questions individually below.
>
> > How does the method ensure robust sensitivity to subtle changes in node and edge features, especially when applied to simpler dynamic graphs or those with more complex interactions beyond temporal events?
>
> Could you please clarify what you mean by "subtle" changes? We believe that our experiments on diversity (Section 4.2) test common failure points of synthetic data while our experiments on fidelity (Section 4.1) test single dimensions of CTDGs. We are happy to provide additional details or consider further evaluations if needed.
> Similarly, could you please elaborate on what you mean by "interactions beyond temporal events"? Our CTDG representation (Equation 1, line 112) consists solely of a timestamped sequence of interactions. This representation is very common in the literature (e.g., [1][2][3]) and generalizes other dynamic graph representations [4].
>
> > In theory, how do graph scale and interaction complexity influence the performance of this metric?
>
> This is a great question! We do not expect the performance of our metric to decrease as a function of scale. When considering scale theoretically, we should examine two dimensions:
>
> 1. __Number of Nodes:__ As the number of nodes increases, the contribution of a perturbation affecting a single node to the overall metric score decreases proportionally. This is because the node representations $\mathbf{v}_j$ (line 272) are normalized, ensuring that each node contributes equally regardless of the total number of nodes.
>
> 2. __Number of Interactions:__ Similarly, as the number of interactions increases, the impact of a single interaction perturbation diminishes relative to the total. The projection matrices $W_1$ and $W_2$ (lines 283 and 291, respectively) are also normalized to preserve distances according to the JL lemma. This normalization maintains consistent sensitivity across different scales.
>
> Our normalization ensures that the effect of individual perturbations scales inversely with the size of the graph. In expectation, the contribution of a single node or interaction perturbation to the overall score decreases linearly with scale. This is a desirable quality. Intuitively, a single change (e.g., edge removal) to a small graph should cause more dissimilarity than a single change to a much larger graph.
>
> >  This paper covers a range of metrics and theoretical concepts. As a minor suggestion, it might enhance clarity to include a high-level figure illustrating the differences between the proposed metric and others, beyond just presenting post-experimental results.
>
> Thanks for your suggestion. It is not immediately clear to us how we can demonstrate the differences between considered metrics using a figure but are definitely open to specific suggestions. We do believe the use of a table may be appropriate to summarize the features of each method. We provide an example of such a table in the next author comment below. If you believe that including this could be helpful to readers, we are happy to add it to the paper.
>
>
>
> [1] Rossi, Emanuele, et al. "Temporal graph networks for deep learning on dynamic graphs." arXiv preprint arXiv:2006.10637(2020).
>
> [2] Jin, Ming, Yuan-Fang Li, and Shirui Pan. "Neural temporal walks: Motif-aware representation learning on continuous-time dynamic graphs." Advances in Neural Information Processing Systems 35 (2022): 19874-19886.
>
> [3] Zhang, Liming, et al. "TG-GAN: Continuous-time temporal graph deep generative models with time-validity constraints." Proceedings of the Web Conference 2021. 2021.
>
> [4] Kazemi, Seyed Mehran. "Dynamic graph neural networks." Graph Neural Networks: Foundations, Frontiers, and Applications (2022): 323-349.

---

> > ### Author Response · Authors · 2024-11-20
> >
> > | Capability                         | Static Metrics | Node Behavior Metrics | Feature Metrics | JL-Metric (Ours) |
> > |------------------------------------|---------------------------|-----------------------|-----------------|------------------|
> > | Unified Single-Value Output        | ✗                         | ✗                     | ✓               | ✓                |
> > | Direct Topology Modeling           | ✓                         | ✓                     | ✗               | ✓                |
> > | Direct Feature Modeling            | ✗                         | ✗                     | ✓               | ✓                |
> > | Captures Temporal Dependencies     | ✗                         | ✓                     | ✗                | ✓                |
> > | Does not Require Static Snapshots  | ✗                         | ✓                     | ✓               | ✓                |
> >
> > Comparison of metric capabilities. Static metrics require multiple measures to evaluate graphs comprehensively, lack sensitivity to interaction features, and require static snapshots. Node behavior metrics capture some temporal dependencies by tracking node-level patterns. Feature metrics model interaction features, assuming i.i.d. data. The JL-Metric unifies these capabilities: it provides a scalar metric that models topology and features while capturing temporal dependencies, all without requiring static snapshot construction.

---

> > > ### Comment · Reviewer_zdbk · 2024-11-22
> > >
> > > Thanks for your response, and the comparison of metric capabilities is very helpful, I would like to raise my score.

---

### Official Review · Reviewer_MPcg · 2024-11-04

**Soundness:** 3
**Presentation:** 3
**Contribution:** 3
**Rating:** 8
**Confidence:** 3

**Summary:**

The primary motivation behind the proposed work is to address the limitations of existing metrics for evaluating generative models for dynamic graphs.
The authors provide various limitation such as: Lack of consideration for temporal dependencies, lacking a unified measure that is sensitive to both features and topology, Absence of a unified scalar metric.

The authors propose Johnson-Lindenstrauss (JL) metric to overcome above limitations.
They leverage the Johnson-Lindenstrauss lemma to project dynamic graphs into a lower-dimensional space. It  allows for comparison of generated and ground-truth graphs using standard distance metrics.


The authors perform evaluation on datasets:  Reddit, Wikipedia, LastFM,
and MOOC.  They compared their proposed JL-Metric with several traditional metrics based on topological and feature-based properties. idelity: Also, they did evaluation w.r.t Diversity,  Sample Efficiency , and Computational Efficiency.
The authors used real-world and synthetic datasets to test the metrics under various conditions, including perturbations like edge rewiring, time perturbation, and event permutation.

**Strengths:**

1. Novel Approach to Evaluating Dynamic Graph Generative Models.

2. Strong Empirical Evaluation.

3. Code is shared.

4. Background work is very well cited and explained. Limitations are clearly highlighted and justified by experiments.

**Weaknesses:**

1. In 4.1 evaluation:

Are all the types of perturbations independent? Can't the perturbations happen jointly? i.e edge rewiring and time perturbation together? Or have I misunderstood it? Is there any assumption. Kindly clarify. If they are independent, can we understand the impact if they occur jointly? Since in reality, it could happen right?

2. "a timestamp ti is replaced by a uniformly selected one trand ∼ Unif(ti−1, ti+1)"
Why is the range so small? just 3 possibilities? Is there any specific reason for this? Can we increase this range while also preserving the order?

**Questions:**

Please see weakness section.

1. Dataset statistics seem to be missing.
"We use a subset of these data (details in Appendix C), which were originally introduced
by Jodie (Kumar et al., 2019) and have become standard CTDG benchmarks"

It is not clear what subset for each dataset? The authors should specify clearly.


Check [A]  Table 1 on what information could be useful to add in terms of dataset statistics.

2. Could authors throw some more light on how evolution is capture in their metric? ". The JL-Metric, by
contrast, is more expressive, capturing both temporal and structural changes directly". Can the authors clarify it better. structural + temporal?
I may be missing something.

[A] TIGGER: Scalable Generative Modelling for Temporal Interaction Graphs
https://aaai.org/papers/06819-tigger-scalable-generative-modelling-for-temporal-interaction-graphs/

---

> ### Author Response · Authors · 2024-11-20
>
> Thanks for your thoughtful and detailed review and for recognizing the novelty of our approach to evaluating dynamic graph generative models, the strength of our empirical evaluation, and the clarity of presentation. We address your comments and questions individually below:
>
> >Are all the types of perturbations independent? Can't the perturbations happen jointly? i.e edge rewiring and time perturbation together? Or have I misunderstood it? Is there any assumption. Kindly clarify. If they are independent, can we understand the impact if they occur jointly? Since in reality, it could happen right?
>
> In our __fidelity__ experiments (Section 4.1; Figure 1, top 3 rows), we independently perturb a single dimension of interest (topology, features, or time), in order to isolate and quantify each perturbation's impact on the metric. We view this as a strength of our sensitivity analysis, as it allows us to precisely determine which types of perturbations each metric is sensitive to. By isolating each dimension, we can identify metrics that may only respond to specific aspects of the graph data. For example, a metric that is sensitive to feature evolution but not to topological changes might appear effective if both perturbations occur simultaneously, potentially masking its limitations.
>
> However, our experiments on __diversity__ (Section 4.2; Figure 2, bottom 2 rows) allow us to study joint perturbations that affect topology, features, and temporal aspects simultaneously. These experiments iteratively modify the graph in all three dimensions. These experiments are designed to represent common failure modes of generative models, as discussed in Section 2.4 (lines 207-219).
>
> > "a timestamp ti is replaced by a uniformly selected one trand ∼ Unif(ti−1, ti+1)" Why is the range so small? just 3 possibilities? Is there any specific reason for this? Can we increase this range while also preserving the order?
>
> We believe this is due to a simple misunderstanding of our sensitivity analysis design for the temporal dimension. The timestamp is uniformly sampled from $t_{\text{rand}} \sim \text{Unif}(t_{i-1}, t_{i+1})$, that is uniformly sampled with support between the previous and next time stamp. It seems you are mistaking this with $t_{\text{rand}} \sim \text{Unif}(t_{i}-1, t_{i}+1)$. For example, assuming integer time resolution, if $t_{i+1}-t_{i-1} = n $, $n \in \mathbb{N}$, then there are $n$ possibilities for the perturbation, not just 3. Increasing the support further will break order preservation and thus lead to a more imprecise test (as described in the response to the previous comment above).
>
> > Dataset statistics seem to be missing. "We use a subset of these data (details in Appendix C), which were originally introduced by Jodie (Kumar et al., 2019) and have become standard CTDG benchmarks"
> It is not clear what subset for each dataset? The authors should specify clearly.
>
> Thanks for the suggestion! Appendix C (lines 825-829) indicates how we selected the subsets for each dataset. We agree that it is a good idea to include dataset statistics and have added a new table (Table 2) to Appendix C. This is similar to the one in TIGGER as you suggest and provides statistics on node count, interaction count, static snapshot count, and event feature cardinality for each dataset.
>
> > Could authors throw some more light on how evolution is capture in their metric? ". The JL-Metric, by contrast, is more expressive, capturing both temporal and structural changes directly". Can the authors clarify it better. structural + temporal? I may be missing something.
>
> The JL-Metric captures evolution through its unified representation of both temporal and topological patterns. Classical metrics like average node degree cannot detect changes in temporal dynamics that preserve the graph's static structure. For example, one could substantially alter the timing of edge formations while maintaining the same degree distribution, leaving this metrics unchanged. The JL-Metric, by directly embedding both temporal information and topological structure into the same space, is sensitive to changes in either aspect. This means it can detect both pure temporal perturbations (e.g., altered interaction timing) and topological changes (e.g., modified connectivity), as well as their combinations.
>
> We believe that part of the confusion may stem from our use of the term "structural" as a synonym for "topological" in the sentence you mention. We have updated the paper to be more clear here.

---

> > ### Comment · Reviewer_MPcg · 2024-11-24
> > **Thanks for update**
> >
> > Thanks for the clarification.and updated manuscript.
> >
> > Good work
> >  I maintain my score.

---

### Comment · Area_Chair_v6mj · 2024-11-28

I would like to encourage the reviewers to engage with the author's replies if they have not already done so. At the very least, please
acknowledge that you have read the rebuttal.

---

### Meta-Review · Area_Chair_v6mj · 2024-12-18

**Metareview:**

The paper proposes the JL-Metric, a novel evaluation metric that use the Johnson-Lindenstrauss lemma to assess dynamic graphs. It addresses some limitations of existing methods (temporal dependencies, topology and feature changes). The authors provide extensive empirical evidence across diverse datasets and perturbation scenarios, including aspects such as fidelity and diversity.

The reviewers praised the efficiency and practicality, the empirical evaluation, and the fact that the metric unifies topology and feature changes. The discussion resulted in several useful suggestions. I encourage the authors to include them in the final version.

**Additional Comments On Reviewer Discussion:**

While some reviewers questioned its novelty, the authors clarified how their work extends prior methods and the challenges that they address (temporal evolution and large-scale evaluation).  The authors provided clarifications on the independence of perturbations, sensitivity to joint changes, and robustness to varying graph scales and complexities. The reviewers agree that the paper warrants acceptance.

---

### Decision · Program_Chairs · 2025-01-22

Accept (Spotlight)